# Lookahead Shielding for Regular Safety Properties in Reinforcement Learning

## Abstract

To deploy reinforcement learning (RL) systems in real-world scenarios we need to consider requirements such as safety and constraint compliance, rather than blindly maximizing for reward. In this paper we develop a lookahead shielding framework for RL with regular safety properties, which on the contrary to prior shielding methodologies requires minimal prior knowledge. At each environment step our framework aims to satisfy the regular safety property for a bounded horizon with high-probability, for the tabular setting we provide provable guarantees. We compare our setup to some common algorithms developed for the constrained Markov decision process (CMDP), and we demonstrate the effectiveness and scalability of our framework by extensively evaluating our framework in both tabular and deep RL environments.

## 1 Introduction

The field of safe reinforcement learning (RL) (García & Fernández, 2015; Amodei et al., 2016) has gained increasing interest, as practitioners begin to understand the challenges of applying RL in the real world (Dulac-Arnold et al., 2019). There exist several distinct paradigms in the literature, including constrained optimization (Chow et al., 2018; Liang et al., 2018; Tessler et al., 2018; Ray et al., 2019; Achiam et al., 2017; Yang et al., 2020), logical constraint satisfaction (Voloshin et al., 2022; Hasanbeig et al., 2018; 2020a;b; De Giacomo et al., 2020; Cai et al., 2021), safety-critical control (McIlvanna et al., 2022; Cheng et al., 2019; Brunke et al., 2022), all of which are unified by prioritizing safety- and risk-awareness during the decision making process.

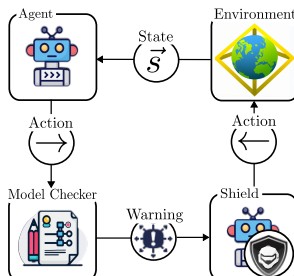

Figure 1: Diagrammatic representation of runtime verification and shielding.

Constrained Markov decision processes (CMDP) (Altman, 1999) have emerged as a popular framework for modelling safe RL, or RL with constraints. Typically, the goal is to obtain a policy that maximizes reward while simultaneously ensuring that the expected cumulative cost remains below a pre-defined threshold. A key limitation of this setting is that constraint violations are enforced in expectation rather than with high probability, the constraint thresholds also have limited semantic meaning, can be very challenging to tune and in some cases inappropriate for highly safety-critical scenarios (Voloshin et al., 2022). Furthermore, the cost function in the CMDP is typically Markovian and thus fails to capture a significantly expressive class of safety properties and constraints.

Regular safety properties (Baier & Katoen, 2008) are interesting because for all but the simplest properties the corresponding cost function is non-Markov. Our problem setup consists of the standard RL objective with regular safety properties as constraints, we note that there has been a significant body of work that combines temporal logic constraints with RL (Voloshin et al., 2022; Hasanbeig et al., 2018; 2020a;b; De Giacomo et al., 2020; Cai et al., 2021), although many of these do not explicitly separate reward and safety in the same way that we do.

Our approach relies on shielding (Alshiekh et al., 2018), which is a safe exploration strategy that ensures the satisfaction of temporal logic constraints by deploying the learned policy in conjunction with a reactive system that overrides any *unsafe* actions. Most shielding approaches typically make highly restrictive assumptions, such as knowledge of the environment dynamics, or at the very least the safety-relevant dynamics of the environment (Alshiekh et al., 2018; Jansen et al., 2020;

Könighofer et al., 2023), full knowledge of the topology of the MDP (Carr et al., 2023), or access to a perfect generative model (Giacobbe et al., 2021), although there has been recent work to relax these restrictions (Goodall & Belardinelli, 2023; He et al., 2022; Xiao et al., 2023). In this paper, we opt for minimal prior knowledge, where the dynamics of the environment are unknown, and runtime verification of the agent is realized by finite horizon model checking with a learned approximation of the environment dynamics. However, in principle our framework is flexible enough to accommodate more standard model checking procedures as long as certain assumptions are met.

Our approach can be summarised as an online shielding approach (see Figure 1), that dynamically identifies unsafe actions during training and deployment, and deploys a learned safe 'backup policy' when necessary. We summarise the main contributions of our paper as follows:

(1) We develop a lookahead shielding framework for RL with regular safety properties as constraints, which requires minimal prior knowledge; unknown transition dynamics and no a priori access to a safe 'backup policy'.

(2) We compare our setup to the CMDP framework and for the tabular setting we provide provable step-wise and episodic safety guarantees, by leveraging ideas such as probabilistic safe sets and irrecoverable actions.

(3) We detail two possible instantiations of our framework for both the tabular and deep RL settings, and we empirically demonstrate the effectiveness of our framework across a variety of environments with different regular safety properties. We compare our approach to projection-based and Lagrange relaxation-based CMDP algorithms.

## 2 PRELIMINARIES

For a finite set $\mathcal{S}$, let $Pow(\mathcal{S})$ denote the power set of $\mathcal{S}$. Also, let $Dist(\mathcal{S})$ denote the set of distributions over $\mathcal{S}$, where a distribution $\mu : \mathcal{S} \to [0,1]$ is a function such that $\sum_{s \in \mathcal{S}} \mu(s) = 1$. Let $\mathcal{S}^*$ and $\mathcal{S}^\omega$ denote the set of finite and infinite sequences over $\mathcal{S}$ respectively. The set of all finite and infinite sequences is denoted $\mathcal{S}^\infty = \mathcal{S}^* \cup \mathcal{S}^\omega$. We denote as $|\rho|$ the length of a sequence $\rho \in \mathcal{S}^\infty$, where $|\rho| = \infty$ if $\rho \in \mathcal{S}^\omega$. We also denote as $\rho[i]$ the $i+1$-th element of a sequence, when $i < |\rho|$, and we denote as $\rho\!\downarrow\, = \rho[|\rho|-1]$ the last element of a sequence, when $\rho \in \mathcal{S}^*$. A sequence $\rho_1$ is a prefix of $\rho_2$, denoted $\rho_1 \preceq \rho_2$, if $|\rho_1| \leq |\rho_2|$ and $\rho_1[i] = \rho_2[i]$ for all $0 \leq i \leq |\rho_1|$. A sequence $\rho_1$ is a proper prefix of $\rho_2$, denoted $\rho_1 \prec \rho_2$, if $\rho_1 \preceq \rho_2$ and $\rho_1 \neq \rho_2$.

**Labelled MDPs and Markov Chains.** An MDP is a tuple $M = (\mathcal{S}, \mathcal{A}, \mathcal{P}, \mathcal{P}_0, \mathcal{R}, AP, L)$, where $\mathcal{S}$ and $\mathcal{A}$ are finite sets of states and actions resp.; $\mathcal{P} : \mathcal{S} \times \mathcal{A} \to Dist(\mathcal{S})$ is the *transition function*; $\mathcal{P}_0 \in Dist(\mathcal{S})$ is the *initial state distribution*; $\mathcal{R} : \mathcal{S} \times \mathcal{A} \to [0,1]$ is the *reward function*; $AP$ is a set of *atomic propositions*, where $\Sigma = Pow(AP)$ is the *alphabet* over $AP$; and $L : \mathcal{S} \to \Sigma$ is a *labelling function*, where $L(s)$ denotes the set of atoms that hold in a given state $s \in \mathcal{S}$. A memory-less (stochastic) *policy* is a function $\pi : \mathcal{S} \to Dist(\mathcal{A})$ and its *value function*, denoted $V_\pi : \mathcal{S} \to \mathbb{R}$ is defined as the *expected discounted reward* from a given state under policy $\pi$, i.e., $V_\pi(s) = \mathbb{E}_\pi[\sum_{t=0}^{T} \gamma^t \mathcal{R}(s_t, a_t)|s_0 = s]$, where $T$ is a fixed episode length and $\gamma$ is the discount factor. Furthermore, denote as $\mathcal{M}_\pi = (\mathcal{S}, \mathcal{P}_\pi, \mathcal{P}_0, AP, L)$ the *Markov chain* induced by a fixed policy $\pi$, where the transition function is such that $\mathcal{P}_\pi(s'|s) = \sum_{a \in A} \mathcal{P}(s'|s,a)\pi(a|s)$. A path $\rho \in \mathcal{S}^\infty$ through $\mathcal{M}_\pi$ is a finite (or infinite) sequence of states. Using standard results from measure theory it can be shown that the set of all paths $\{\rho \in \mathcal{S}^\omega \mid \rho_{pref} \preceq \rho\}$ with a common prefix $\rho_{pref}$ is measurable (Baier & Katoen, 2008).

**Probabilistic CTL.** (PCTL) (Baier & Katoen, 2008) is a branching-time temporal logic for specifying properties of stochastic systems. A well-formed PCTL property can be constructed with the following grammar,

$$\Phi ::= \text{true} \mid a \mid \neg\Phi \mid \Phi \wedge \Phi \mid \mathbb{P}_{\bowtie p}[\varphi]$$

$$\varphi ::= X\Phi \mid \Phi U \Phi \mid \Phi U^{\leq n} \Phi$$

where $a \in AP$, $\bowtie \in \{<, >, \leq, \geq\}$ is a binary comparison operator, and $p \in [0,1]$ is a probability. Negation $\neg$ and conjunction $\wedge$ are the familiar logical operators from propositional logic, and next $X$, until $U$ and bounded until $U^{\leq n}$ are the temporal operators from CTL (Baier & Katoen, 2008). We make the distinction here between state formula $\Phi$ and path formula $\varphi$. The satisfaction relation for state formula $\Phi$ is defined in the standard way for Boolean connectives. For probabilistic

quantification we say that $s \models \mathbb{P}_{\bowtie p}[\varphi]$ iff $\Pr(s \models \varphi) := \Pr(\rho \in S^\omega \mid \rho[0] = s, \rho \models \varphi) \bowtie p$. Let $\Pr_{\mathcal{M}}(s \models \varphi)$ be the probability w.r.t. the Markov chain $\mathcal{M}$. For path formula $\varphi$ the satisfaction relation is also defined in the standard way for temporal logics, see Baier & Katoen (2008) . We also note that the important temporal operators 'eventually' $\Diamond$ and 'always' $\Box$, and their bounded counterparts $\Diamond^{\leq n}$ and $\Box^{\leq n}$ can be derived in a familiar way, i.e., $\Diamond \, \Phi ::= \text{true}\, U \Phi, \Box \, \Phi ::= \neg \Diamond \neg \Phi$, resp. $\Diamond^{\leq n} \, \Phi ::= \text{true}\, U^{\leq n}\Phi, \Box^{\leq n} \, \Phi ::= \neg \Diamond^{\leq n} \neg \Phi$.

**Regular Safety Property.** A linear time property $P_{safe} \subseteq \Sigma^\omega$ over the alphabet $\Sigma$ is a safety property if for all words $w \in \Sigma^\omega \setminus P_{safe}$, there exists a finite prefix $w_{pref}$ of $w$ such that $P_{safe} \cap \{w' \in \Sigma^\omega \mid w_{pref} \preceq w'\} = \varnothing$. Any such sequence $w_{pref}$ is called a *bad prefix* for $P_{safe}$, a bad prefix $w_{pref}$ is called *minimal* iff there does not exist $w'' \prec w_{pref}$ such that $w''$ is a bad prefix for $P_{safe}$. Let *BadPref*$(P_{safe})$ and *MinBadPref*$(P_{safe})$ denote the set of of bad and minimal bad prefixes resp.

A safety property $P_{safe} \in \Sigma^\omega$ is *regular* if the set *BadPref*$(P_{safe})$ constitutes a regular language. That is, there exists some *deterministic finite automata* (DFA) that accepts the bad prefixes for $P_{safe}$ (Baier & Katoen, 2008), that is, a path $\rho \in \mathcal{S}^\omega$ is 'unsafe' if the trace $trace(\rho) = L(\rho[0]), L(\rho[1]), \ldots \in \Sigma^\omega$ is accepted by the corresponding DFA.

**Definition 2.1** (DFA). *A deterministic finite automata is a tuple $\mathcal{D} = (\mathcal{Q}, \Sigma, \Delta, \mathcal{Q}_0, \mathcal{F})$, where $\mathcal{Q}$ is a finite set of states, $\Sigma$ is a finite alphabet, $\Delta : \mathcal{Q} \times \Sigma \to \mathcal{Q}$ is the transition function, $\mathcal{Q}_0$ is the initial state, and $\mathcal{F} \subseteq \mathcal{Q}$ is the set of accepting states. The extended transition function $\Delta^*$ is the total function $\Delta^* : \mathcal{Q} \times \Sigma^* \to \mathcal{Q}$ defined recursively as $\Delta^*(q, w) = \Delta(\Delta^*(q, w \setminus w\downarrow), w\downarrow)$. The language accepted by DFA $\mathcal{D}$ is denoted $\mathcal{L}(\mathcal{D}) = \{w \in \Sigma^* \mid \Delta^*(\mathcal{Q}_0, w) \in \mathcal{F}\}$.*

Furthermore, we denote $P_{safe}^N \subseteq \Sigma^\omega$ as the corresponding finite-horizon safety property for $N \in \mathbb{Z}_+$, where for all words $w \in \Sigma^\omega \setminus P_{safe}^N$ there exists $w_{pref} \preceq w$ such that $|w_{pref}| \leq N$ and $w_{pref} \in$ *BadPref*$(P_{safe})$. We model check regular safety properties by synchronizing the DFA and Markov chain in a standard way, by computing the product Markov chain.

**Definition 2.2** (Product Markov Chain). *Let $\mathcal{M} = (\mathcal{S}, \mathcal{P}, \mathcal{P}_0, AP, L)$ be a Markov chain and $\mathcal{D} = (\mathcal{Q}, \Sigma, \Delta, \mathcal{Q}_0, \mathcal{F})$ be a DFA. The product Markov chain is $\mathcal{M} \otimes \mathcal{D} = (\mathcal{S} \times \mathcal{Q}, \mathcal{P}', \mathcal{P}'_0, \{accept\}, L')$, where $L'(\langle s, q\rangle) = \{accept\}$ if $q \in \mathcal{F}$ and $L'(\langle s, q\rangle) = \varnothing$ o/w, $\mathcal{P}'_0(\langle s, q\rangle) = \mathcal{P}_0(s)$ if $q = \Delta(\mathcal{Q}_0, L(s))$ and $0$ o/w, and $\mathcal{P}'(\langle s', q'\rangle | \langle s, q\rangle) = \mathcal{P}(s'|s)$ if $q' = \Delta(q, L(s'))$ and $0$ o/w.*

**Definition 2.3** (Satisfaction probability for $P_{safe}$). *Let $\mathcal{M} = (\mathcal{S}, \mathcal{P}, \mathcal{P}_0, AP, L)$ be a Markov chain and let $\mathcal{D} = (\mathcal{Q}, \Sigma, \Delta, \mathcal{Q}_0, \mathcal{F})$ be the DFA such that $\mathcal{L}(\mathcal{D}) = BadPref(P_{safe})$. For a path $\rho \in \mathcal{S}^\omega$ in the Markov chain, let $trace(\rho) = L(\rho[0]), L(\rho[1]), \ldots \in \Sigma^\omega$ be the corresponding word over $\Sigma = Pow(AP)$. From a given state $s \in \mathcal{S}$ the satisfaction probability for $P_{safe}$ is defined as follows,*

$$\Pr_{\mathcal{M}}(s \models P_{safe}) := \Pr_{\mathcal{M}}(\rho \in \mathcal{S}^\omega \mid \rho[0] = s, trace(\rho) \notin \mathcal{L}(\mathcal{D}))$$

*Perhaps more importantly, we note that this satisfaction probability can be written as the following reachability probability in the product Markov chain,*

$$\Pr_{\mathcal{M}}(s \models P_{safe}) = \Pr_{\mathcal{M} \otimes \mathcal{D}}(\langle s, q_s\rangle \not\models \Diamond accept)$$

*where $q_s = \Delta(\mathcal{Q}_0, L(s))$ and $\Diamond accept$ is a probabilistic CTL path formula that reads, 'eventually accept' (Baier & Katoen, 2008).*

The finite-horizon satisfaction probability of $P_{safe}$ can be equated to the to the satisfaction probability of the corresponding finite horizon safety property $P_{safe}^N$ as follows.

**Proposition 2.4** (Finite-horizon satisfaction probability for $P_{safe}$). *Let $\mathcal{M}$ and $\mathcal{D}$ be defined as in Defn. 2.3. For a path $\rho \in \mathcal{S}^\omega$, let $trace_N(\rho) = L(\rho[0]), L(\rho[1]) \ldots, L(\rho[N])$ be the corresponding finite word over $\Sigma = Pow(AP)$. For a given state $s \in \mathcal{S}$ the finite horizon satisfaction probability for $P_{safe}$ is given by,*

$$\Pr_{\mathcal{M}}(s \models P_{safe}^N) := \Pr_{\mathcal{M}}(\rho \in \mathcal{S}^\omega \mid \rho[0] = s, trace_N(\rho) \notin \mathcal{L}(\mathcal{D}))$$

*where $N \in \mathbb{Z}_+$ is some fixed model checking horizon. Similar to before, we show that the finite horizon satisfaction probability can be written as the following bounded reachability probability,*

$$\Pr_{\mathcal{M}}(s \models P_{safe}^N) = \Pr_{\mathcal{M} \otimes \mathcal{D}}(\langle s, q_s\rangle \not\models \Diamond^{\leq N} accept)$$

*where $q_s = \Delta(\mathcal{Q}_0, L(s))$ is as before and $\Diamond^{\leq N} accept$ is the corresponding step-bounded probabilistic CTL path formula that reads, 'eventually accept in N timesteps'.*

## 3 LOOKAHEAD SHIELDING

Our goal is to synthesize a safe policy, $\pi_{sh} : \mathcal{S} \times \mathcal{Q} \rightarrow Dist(\mathcal{A})$, by dynamically integrating two sub-policies, the 'task policy', denoted $\pi_r : \mathcal{S} \rightarrow Dist(\mathcal{A})$ and the 'backup policy' $\pi_b : \mathcal{S} \times \mathcal{Q} \rightarrow Dist(\mathcal{A})$. Control of the agent is given to one of these sub-policies depending on the current state of the agent and the desired safety-threshold. The 'task policy' $\pi_r$ is trained with RL to maximize reward, i.e., $\max_\pi V_\pi$. On the other hand the 'backup policy' $\pi_b$ is (usually) a low-reward policy, specifically designed to keep the agent within a *probabilistic safe set* of states. In some simple instances the 'backup policy' may constitute a simple rule-based policy that is guaranteed to be safe before training. However, since we assume minimal prior knowledge, the 'backup policy' will need to be trained online with RL similar to the 'task policy'.

From a given product state $\langle s, q \rangle \in \mathcal{S} \times \mathcal{Q}$, we dynamically switch between $\pi_r$ and $\pi_b$ by evaluating the $N$-step conditional action reachability probability, defined as follows,

**Definition 3.1** ($N$-step conditional action reachability probability). *Let $a \in \mathcal{A}$ be an action, let $M = (\mathcal{S}, \mathcal{A}, \mathcal{P}, \mathcal{P}_0, \mathcal{R}, AP, L)$ be an MDP and let $\mathcal{D} = (\mathcal{Q}, \Sigma, \Delta, \mathcal{Q}_0, \mathcal{F})$ be the DFA such that $\mathcal{L}(\mathcal{D}) = BadPref(P_{safe})$. The $N$-step conditional action reachability probability, is computed from the MDP by first resolving the non-determinism of the MDP with action $a$ and then with $\pi_b$ thereafter, this is denoted, $\mathrm{Pr}_{\pi_b}^a(\langle s, q \rangle \models \Diamond^{\leq N} accept)$.* [1]

In our case, we estimate the $N$-step conditional action reachability probability $\mathrm{Pr}_{\pi_b}^a(\langle s, q \rangle \models \Diamond^{\leq N} accept)$ by rolling-out a learned dynamics model for $N$ timesteps. If this probability does not exceed some step-wise safety-threshold $\varepsilon_t$ then the action $a$ is permissible, otherwise the action $a$ is rejected and a safe action $a'$ is sampled from the $\pi_b$. Thus, the 'shielded policy' $\pi_{sh}$ has the following form,

$$\pi_{sh}(\langle s, q \rangle, a) = \begin{cases} \pi_r(s, a) & \text{if } \mathrm{Pr}_{\pi_b}^a(\langle s, q \rangle \models \Diamond^{\leq N} accept) \leq \varepsilon_t \\ \pi_b(\langle s, q \rangle, a) & \text{otherwise} \end{cases} \tag{1}$$

The safety of $\pi_{sh}$ relies on the fact that the 'backup policy' $\pi_b$ can keep the agent within a probabilistic safe set, and that for any *irrecoverable* action $a$ the lookahead or model checking horizon $N$ is sufficiently large. We will formalize both these ideas later on in Section 3.5.

Thus $\pi_{sh}$ provides a step-wise safety guarantee of $\varepsilon_t$ which is in line with similar shielding approaches (Wabersich et al., 2021; Bastani et al., 2021). For the satisfaction of $P_{safe}$ for an entire fixed episode length $T$, we can use a conservative union bound to derive a probability lower bound, $\mathrm{Pr}_{\mathcal{M}}(s \models P_{safe}) \geq 1 - \varepsilon$ or equivalently, $\mathrm{Pr}_{\mathcal{M} \otimes \mathcal{D}}(\langle s, q_s \rangle \models \Diamond accept) \leq \varepsilon$, where $\varepsilon = \sum_{t=0}^{T} \varepsilon_t$. Unfortunately, we cannot immediately derive an infinite horizon guarantee, without for example, either assuming the existence of and being able to identify *safe end components* (Haddad & Monmege, 2018; Brázdil et al., 2024), or assuming deterministic dynamics (Berkenkamp et al., 2017).

### 3.1 TRAINING THE BACKUP POLICY

As we alluded to above, in all but the simplest cases the 'backup policy' $\pi_b$ will need to be trained online with RL. To construct an effective 'backup policy' we introduce the following cost function,

**Definition 3.2** (Cost function). *Let $P_{safe}$ be a regular safety property and let $\mathcal{D}$ be the DFA such that $\mathcal{L}(\mathcal{D}) = BadPref(P_{safe})$, the cost function is an $\omega$-automaton (or Büchi automaton) that simulates the DFA $\mathcal{D}$ and then resets after reaching an accepting state (i.e. for all $q \in \mathcal{F}$, $q \rightarrow \mathcal{Q}_0$), the cost function $\mathcal{C}$ is then defined as follows:*

$$\mathcal{C}(\langle s, q \rangle) = \begin{cases} 1 & \text{if } accept \in L'(\langle s, q \rangle) \\ 0 & \text{otherwise} \end{cases}$$

*where $L'$ is the labelling function defined in Definition 2.2.*

The 'backup policy' $\pi_b$ can then be trained with standard RL techniques (e.g. Q-learning) to the minimize the *expected discounted cost*, i.e. $\mathbb{E}_\pi[\sum_{t=0}^{\infty} \gamma^t \mathcal{C}(s_t, q_t)]$.

---

[1] The probability here is taken under the product $\mathcal{M}_{\pi_b} \otimes \mathcal{D}$ with the first timestep replaced by the conditional action matrix $\boldsymbol{P}^{(a)}$, this value is well-defined and can be computed exactly (see Algorithm 3).

**Remark 3.3.** *It is important to note that for regular safety properties the corresponding cost function is defined over the product states and is thus non-Markov. As a result the 'backup policy' is also defined over the product states, which can pose an issue, particularly for larger automata, as the rate of convergence will be much slower than expected. To eliminate this issue we leverage* counterfactual experiences *(CFE) (Icarte et al., 2022; 2018) – a method originally used for reward machines which generates additional experience for the policy, by simulating automaton transitions.*

## 3.2 COMPARISON TO CONSTRAINED MDP

We now provide a comparison to the CMDP framework (Altman, 1999; Ray et al., 2019), where typically the constraints are specified as expected cumulative cost constraints at the trajectory level.

**Problem 3.4** (Expected Cumulative Cost Constraint)**.**

$$\max_\pi V_\pi \quad subject\ to \quad \mathbb{E}\left[\sum_{t=0}^T \mathcal{C}(\langle s_t, q_t\rangle)\right] \le C \tag{2}$$

*where* $\mathcal{C} : \mathcal{S} \times \mathcal{Q} \to \mathbb{R}$ *is the cost function from Definition 3.2 and* $C > 0$ *is the cost threshold.*

To guarantee the satisfaction of $P_{safe}$ with probability at least $1-\varepsilon$ for the entire fixed episode length $T$, the cost threshold $C$ needs to be set to a prohibitively small value (namely $\varepsilon$), which algorithms developed to tackle CMDPs, like PPO-Lagrangian (PPO-Lag) (Ray et al., 2019) and Constrained Policy Optimization (CPO) (Achiam et al., 2017) are not always suited for.

## 3.3 TABULAR RL

For tabular RL it is most natural to use tabular Q-learning (QL) for training both the 'task policy' and 'backup policy'. The update rule for the 'task policy' $\pi_r$ is modified slightly, to give zero reward to actions that are not permissible,

$$\hat{Q}_r(s_t, a_t) \overset{\alpha}{\longleftarrow} \begin{cases} R(s_t, a_t) + \gamma \max_a\{\hat{Q}_r(s_{t+1}, a)\} & \text{if } \Pr_{\pi_b}^{a_t}(\langle s, q\rangle \models \Diamond^{\le N} accept) \le \varepsilon_t \\ 0 & \text{otherwise} \end{cases} \tag{3}$$

where $\overset{\alpha}{\longleftarrow}$ denotes an in-place update with learning rate $\alpha$. This modification prevents the shielded policy from 'getting stuck' proposing possibly high-reward but unsafe actions and should reduce the number of times the 'task policy' is overridden. The 'backup policy' $\pi_b$ is updated with the standard QL update rule, but with penalties supplied by the cost function from Definition 3.2,

$$\hat{Q}_b(s_t, q_t, a_t) \overset{\alpha}{\longleftarrow} \gamma \max_a\{\hat{Q}_b(s_{t+1}, q_{t+1}, a)\} - \mathcal{C}((s_t, q_t)) \tag{4}$$

For dynamics learning, we estimate the transition probabilities by using the empirical transition probabilities $\widehat{\mathcal{P}}(s' \mid s, a) = \#(s', s, a)/\#(s, a)$, where $\#(s, a)$ and $\#(s', s, a)$ are the visit counts for $(s, a)$ and $(s', s, a)$ respectively. The full algorithm is detailed in Appendix A.1.

### 3.3.1 DEEP RL

For our deep RL experiments we use DreamerV3 (Hafner et al., 2023) for both dynamics learning and policy optimization. DreamerV3 is based on the *Recurrent State Space Model* (RSSM) (Hafner et al., 2019), a special type of sequential *Variational Auto-encoder* (VAE) (Kingma & Welling, 2013), which learns a latent representation and dynamics model of the environment from observations. The model consists of the following key components: sequential model $h_t = f_\theta(h_{t-1}, z_{t-1}, a_{t-1})$, observation encoder $z_t \sim q_\theta(z_t \mid o_t, h_t)$, transition predictor $\hat{z}_t \sim p_\theta(\hat{z}_t \mid h_t)$, observation decoder $\hat{o}_t \sim p_\theta(\hat{o}_t \mid h_t, z_t)$, reward predictor $\hat{r}_t \sim p_\theta(\hat{r}_t \mid h_t, z_t)$ and termination predictor $\hat{\gamma}_t \sim p_\theta(\hat{\gamma}_t \mid h_t, z_t)$. Our implementation is build upon approximate model-based shielding (AMBS) (Goodall & Belardinelli, 2023) which additionally uses a cost predictor $\hat{c}_t \sim p_\theta(\hat{c}_t \mid h_t, z_t)$ to predict state-dependent costs. Since DreamerV3 encodes the observation and action history in the latent vectors $(h_t, z_t)$ we can use the same cost predictor to learn the cost function $\mathcal{C}(\langle s_t, q_t\rangle)$ from Definition 3.2, we back-propagate the cost predictor gradients through the RSSM, with the hope that the necessary temporal dependencies for predicting the cost are captured in the latent space. We can then estimate $\Pr_{\pi_b}^a(\langle s, q\rangle \models \Diamond^{\le N} accept)$ by rolling out the latent dynamics model $p_\theta$, summing the predicted costs along the trajectories and average the result over multiple trajectories sampled in parallel. The full algorithm is detailed in Appendix A.3.

### 3.4 Model Checking

We now detail several model checking paradigms that can be 'plugged' into our framework for computing the finite-horizon satisfaction probability of the regular safety property $P_{safe}$.

**Exact model checking.** If we have access to the transition matrix $\mathcal{P}$ of the MDP then we can exactly compute the (finite horizon) satisfaction probability of $P_{safe}$, in the Markov chain $\mathcal{M}_\pi$ induced by the fixed policy $\pi$ in time $\mathcal{O}(\text{poly}(\text{size}(\mathcal{M}_\pi \otimes \mathcal{D})) \cdot N)$ (Baier & Katoen, 2008) by $O(N)$ matrix multiplications, where $\mathcal{D}$ is the DFA such that $\mathcal{L}(\mathcal{D}) = BadPref(P_{safe})$ and $N$ is the model checking horizon. If the size of the product $\mathcal{M}_\pi \otimes \mathcal{D}$ is too large then exact model checking is impractical.

**Statistical model checking.** To address the limitations of exact model checking, we can construct an estimate of $\text{Pr}_{\pi_b}^a(\langle s, q \rangle \models \Diamond^{\leq N} accept)$ by computing the proportion of accepting paths from a set of samples generated using the transition matrix of the MDP $\mathcal{P}$. Using statistical bounds, such as Hoeffding's inequality (Hoeffding, 1963) or Bernstein-type bounds (Maurer & Pontil, 2009), we can bound the error of this estimate, with high probability. Since the product states $\langle s, q \rangle \in \mathcal{S} \times \mathcal{Q}$ can be computed *on-the-fly*, rather, the time complexity of this approach depends on the horizon $N$, the desired level of accuracy $\varepsilon'$ and failure probability $\delta'$.

**Proposition 3.5.** *Let $\varepsilon' > 0$, $\delta' > 0$, $\langle s, q \rangle \in \mathcal{S} \times \mathcal{Q}$ and $N \geq 1$ be given. By sampling $m \geq \frac{1}{2\varepsilon'^2} \log\left(\frac{2}{\delta'}\right)$ many paths with $\mathcal{P}$, we can obtain an $\varepsilon'$-approximate estimate for the probability $\text{Pr}_{\pi_b}^a(\langle s, q \rangle \models \Diamond^{\leq N} accept)$ with probability at least $1 - \delta'$.*

**Model checking with approximate models.** In the standard RL setting where the transition matrix $\mathcal{P}$ is unknown we can instead rely on an empirical estimate of $\mathcal{P}$ or an 'approximate model', which can either be constructed ahead of time (offline) or from the experience collected during training. We can then either exact model check with the empirical probabilities $\widehat{\mathcal{P}}$, or if the product MC is too large, we can leverage statistical model checking by sampling paths from the 'approximate model'.

**Proposition 3.6.** *Let $\varepsilon' > 0$, $\delta' > 0$, $s \in \mathcal{S}$ and $N \geq 1$ be given. Suppose that for all $(s, a) \in \mathcal{S} \times \mathcal{A}$, our empirical estimate $\widehat{\mathcal{P}}$ is such that,*

$$D_{TV}\left(\mathcal{P}(\cdot \mid s, a), \widehat{\mathcal{P}}(\cdot \mid s, a)\right) \leq \varepsilon'/N \tag{5}$$

*where $D_{TV}$ denotes the total variation (TV) distance [2], then,*

*(1) We can obtain an $\varepsilon'$-approximate estimate for $\text{Pr}_{\pi_b}^a(\langle s, q \rangle \models \Diamond^{\leq N} accept)$ with probability 1 by exact model checking with the transition probabilities of $\widehat{\mathcal{P}}$ in time $\mathcal{O}(\text{poly}(\text{size}(\mathcal{M}_\pi \otimes \mathcal{D})) \cdot N)$.*

*(2) We can obtain an $\varepsilon'$-approximate estimate for $\text{Pr}_{\pi_b}^a(\langle s, q \rangle \models \Diamond^{\leq N} accept)$ with probability at least $1 - \delta'$, by sampling $m \geq \frac{2}{\varepsilon'^2} \log\left(\frac{2}{\delta'}\right)$ many paths with the 'approximate model' $\widehat{\mathcal{P}}$.*

It might be interesting to analyze when (5) is satisfied in practice. For the tabular case we provide this analysis in the proof of Theorem 3.11, stated in the next section. For the deep RL setting, it becomes very tricky to obtain any guarantees, although we can fall back on the upper bound and intuition provided by Goodall & Belardinelli (2023).

### 3.5 Global safety guarantees

In the tabular setting (see Section 3.3) we can prove that $\pi_{sh}$ provides a step-wise safety guarantee of $\varepsilon_t$. We first provide the following definitions.

**Definition 3.7** (Probabilistic Safe Set). *For a given policy $\pi$ defined over the product state space $\mathcal{S} \times \mathcal{Q}$, a probabilistic safe set for the fixed episode length $T$ and step-wise safety level $\varepsilon_t$ is defined,*

$$\mathcal{S}^\pi(\varepsilon_t) = \{\langle s, q \rangle \in \mathcal{S} \times \mathcal{Q} : \text{Pr}_{\mathcal{M}_\pi \otimes \mathcal{D}}(\langle s, q \rangle \models \Diamond^{\leq T} accept) \leq \varepsilon_t\} \tag{6}$$

**Definition 3.8** (Irrecoverable). *An action $a$ is said to be irrecoverable from a given product state $\langle s, q \rangle \in \mathcal{S} \times \mathcal{Q}$, if given $a$ then $\langle s, q \rangle \notin \mathcal{S}^{\pi_b}(\varepsilon_t)$, or in words, $a$ is irrecoverable from $\langle s, q \rangle$ if given $a$ the product state $\langle s, q \rangle$ is not in the (T-step) probabilistic safe set for the 'backup policy' $\pi_b$.*

---

[2] For two discrete probability distributions $\mu_1$ and $\mu_2$ over the same space $\mathcal{X}$ the TV distance is defined as: $D_{TV}(\mu_1(\cdot), \mu_2(\cdot)) = \frac{1}{2} \sum_{x \in X} |\mu_1(x) - \mu_2(x)|$

Ideas such as *probabilistic safe sets* and *irrecoverable* states/actions have been considered in many prior works (Abate et al., 2008; Hewing & Zeilinger, 2018; Li & Bastani, 2020; Bastani et al., 2021; Thomas et al., 2021). Intuitively, the 'backup policy' $\pi_b$ is defined by the ($T$-step) probabilistic safe set from which we can obtain a step-wise safety guarantee of $\varepsilon_t$ (by using the 'backup policy'). Thus, any action $a \in \mathcal{A}$ which does not keep us within this probabilistic safe set is deemed 'irrecoverable'. To complete our proof we need to make the following assumptions.

**Assumption 3.9.** *There exists some $N^* \ll T$ such that for all irrecoverable actions $a \in \mathcal{A}$ the conditional action probability $\Pr_{\pi_b}^a(\langle s, q \rangle \models \Diamond^{\leq N^*} \text{accept})$ and we have chosen $N \geq N^*$.*

**Assumption 3.10.** *The initial state $\langle s_0, L(s_0) \rangle$ is contained in the probabilistic safe set $\mathcal{S}^{\pi_b}(\varepsilon_t)$.*

Assumption 3.9 is for practical convenience, a similar assumption was made in Thomas et al. (2021), it means we can identify irrecoverable actions by only model checking with some fixed horizon $N \geq N^*$, rather than for the entire episode length $T$, which could be either computationally expensive or incur significant model drift when using the empirical estimates of the transition probabilities. Assumption 3.10 guarantees that there is a safe strategy from the initial state, this allows us to prove safety by establishing an invariant: 'we can always fall back on the backup policy for a step-wise safety guarantee of $\varepsilon_t$ regardless of the previous action'.

In general it is unlikely that Assumption 3.10 and 3.9 are immediately satisfied at the start of training, however by using RL to train $\pi_b$ online with penalties provided by the cost function we might expect $\pi_b$ to converge to a policy satisfying these assumptions. Abate et al. (2008) analyse the conditions for the existence of a *maximally safe policy* trained solely with a cost function, this is beyond the scope of our paper, we simply assume that $\pi_b$ satisfies Assumption 3.10 and 3.9 without necessarily being maximally safe.

**Theorem 3.11.** *Under Assumption 3.9 and 3.10, and provided that every state action pair $(s, a) \in \mathcal{S} \times \mathcal{A}$ has been visited at least $\mathcal{O}\left( \frac{N^2 |\mathcal{S}|}{\varepsilon'^2} \log \left( \frac{|\mathcal{A}||\mathcal{S}|}{\delta'} \right) \right)$ times. Then the 'shielded policy' $\pi_{sh}$ provides a step-wise safety guarantee of $\varepsilon_t$ and with a step-wise failure probability of $\delta_t = 2\delta'$.*

The theory is quite conservative here due to the strong dependence on $|S|$, in practice the outermost $|\mathcal{S}|$ can be replaced by the maximum number of successor states $k$ from any given state. Similar to before, by taking a conservative union bound, we can obtain an 'episodic' safety guarantee of $\Pr_{\mathcal{M}}(s \models P_{safe}) \geq 1 - \varepsilon$ with probability $1 - \delta$, where $\varepsilon = \sum_{t=0}^{T} \varepsilon_t$ and $\delta = \sum_{t=0}^{T} \delta_t$.

## 4 EXPERIMENTAL EVALUATION

### 4.1 TABULAR RL

We evaluate our framework in 4 separate tabular environments, see Figure 2. We compare our approach to tabular QL, tabular QL with penalties provided by the cost function in Definition 3.2 (QL-Cost), and two CMDP-based approaches PPO-Lag (Ray et al., 2019) and CPO (Achiam et al., 2017). This instantiation of our framework is called QL-Shield and is detailed in Section 3.3, for model checking we use statistical model checking and we assume no knowledge of the transition matrix $\mathcal{P}$. We briefly summarize the environments here, however, the full environment descriptions can be found in Appendix C.

Inspired by Bura et al. (2022), the *Media Streaming* environment is a simple environment with 20 state and 2 actions. The agent is tasked with managing a data-buffer and the safety property is a simple invariant property: $\Box \neg empty$. Inspired by Hasanbeig et al. (2020a), we test our approach on a sparse *Bridge Crossing* environment, the agent operates in a 'slippery' gridworld, the goal is to reach the opposite side of the bride, the safety property is a simple invariant property: $\Box \neg red$. We test our approach on two more 'slippery' gridworlds,

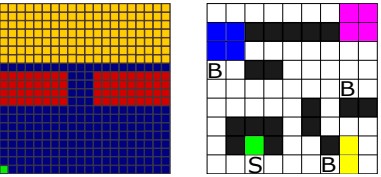

(a) Bridge crossing    (b) $9 \times 9$ gridworld

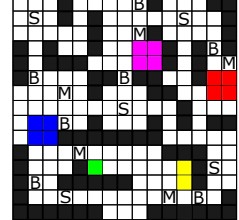

(c) $15 \times 15$ gridworld

Figure 2: Gridworld Environments

a $9 \times 9$ *gridworld* and a $15 \times 15$ *gridworld* with unsafe 'bomb' states, we specify the following three properties in these environments and test them independently, the first (1) is a simple invariant property $\Box \neg B$, the second (2) is $\Box((\neg BXB) \to (XB))$. In words, (1) specifies that the agent must avoid 'bomb' states ($B$), (2) specifies that the agent must 'disarm' 'bomb' states ($B$) by staying on them for at least 2 timesteps. The third (3) is again more complex, specifying that if a 'bomb' state is reached the agent must navigate to a 'medic' state within 10 timesteps and stay there for at least 2 timesteps, this is denoted as $\Box(B \to \Diamond^{\leq 10} \Box^{\leq 2} M)$.

## 4.2 DEEP RL

We evaluate our framework on *Atari Seaquest*, provided as part of the Arcade Learning Environment (ALE) (Machado et al., 2018). Our approach in this setting is built upon DreamerV3 (Hafner et al., 2023), see Section 3.3.1 for details. We compare our approach to vanilla DreamerV3 (no costs), a modified version of DreamerV3 that implements the Augmented Lagrangian (Wright, 2006) very similar in principle to other works such as Safe-DreamerV3 (Huang et al., 2023) and LAMBDA (As et al., 2022), for a detailed description of the Augmented Lagrangian framework we refer the reader to Appendix E.2. We also run PPO-Lag (Ray et al., 2019) and CPO (Achiam et al., 2017) in this setting, however since both these algorithms are model-free and also not suitably adapted to pixel input, we provide as input, perfect RAM information [3] and the current automaton state, this circumvents the issue of PPO-Lag and CPO having to learn an image feature representation and provides a more fair comparison.

Atari Seaquest (see Fig. 3) is a partially observable environment meaning, we do not have direct access to the underlying state space $\mathcal{S}$, we are however provided with observations $o \in O$ as pixel images which correspond to $64 \times 64 \times 3$ tensors. Fortunately DreamerV3 is specifically designed to operate such settings. The cardinality of the action space is $|\mathcal{A}| = 18$. We experiment with two different regular safety properties in this environment, (1) ($\Box \neg surface \to \Box(surface \to diver)) \wedge (\Box \neg out\text{-}of\text{-}oxygen) \wedge (\Box \neg hit)$, and (2) $\Box diver \wedge \neg surface \to \Diamond^{\leq 30} surface$. The first property (1) is aligned closely with the goal – the agent must only surface with

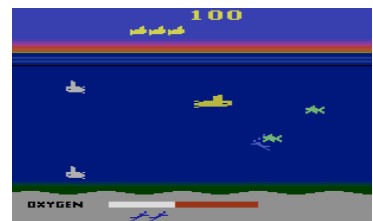

Figure 3: Atari Seaquest. The goal is to rescue divers (*small blue people*), while shooting enemy *sharks* and *submarines*.

a diver, not run out of oxygen and not be hit by an enemy. The second property (2) states after the agent picks up a diver it must return to the surface within 30 timesteps, this property directly conflicts with the optimal policy. For more details we refer the reader to Appendix C.

## 4.3 PRACTICAL CONSIDERATIONS AND LIMITATIONS

A practical comparison between our approach and LTL-based approaches from the literature becomes challenging, as many approaches do not separate reward and safety in to two distinct objectives (Hasanbeig et al., 2018; De Giacomo et al., 2020; Cai et al., 2021), those that do either assume access to a perfect generative model that can be sampled from any state-action pair Voloshin et al. (2022), or assume knowledge of the optimal discount factor $\gamma$ and dual variable $\lambda$ (Shah et al., 2024; Voloshin et al., 2023). LTL and regular safety constraints are slightly different and care would also need to be taken to convert the satisfaction condition for Büchi automata to regular safety properties (or vice versa). In some instances Shah et al., 2024; Voloshin et al., 2023, an unconstrained objective with dual variable $\lambda$ is maximized, which is, for the most part, comparable to our baseline QL-Cost, which also treats the weighting of the cost function as a hyperparameter.

Our approach, is not without its limitations, in particular choosing the model checking horizon $N$ is imperative for safety performance. In principle, any $N = T$ should suffice for episodic guarantees, however large $N$ can incur significant overhead at each decision making step, due repeated model checking. Assumption 3.9 tells us that $N \geq N^*$ is enough, but there is not a practical way choosing $N$ without knowing something about the environment, e.g. for Atari Seaquest it takes at least 25 timesteps, to reach the surface from the bottom of the sea, so $N = 30$ was chosen.

---

[3] The perfect RAM input $x$ corresponds to the features identified in (Anand et al., 2019) and the one-step deltas $\Delta x$ which encodes the necessary temporal information for effective learning.

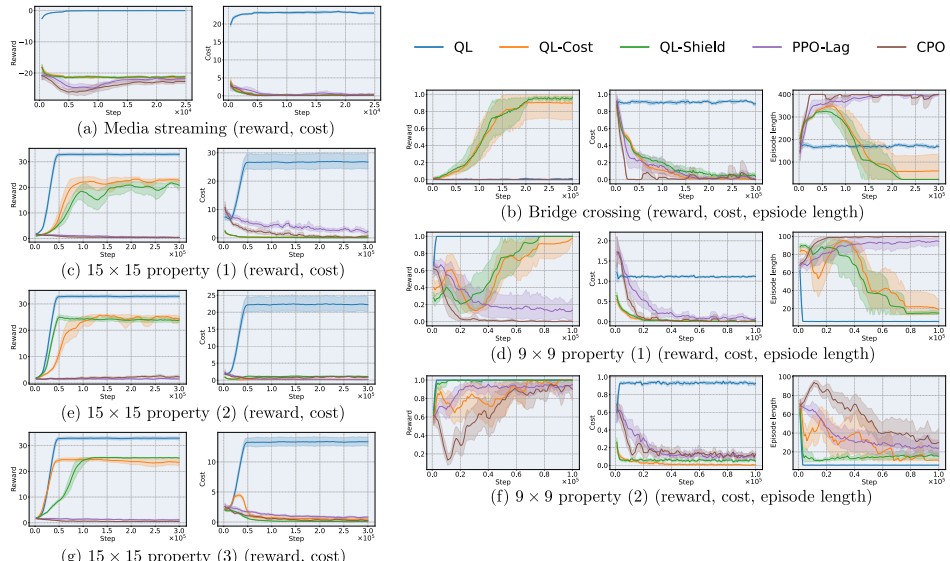

Figure 4: Learning curves for tabular gridworld environments.

## 4.4 DISCUSSION

The media streaming environment is more of a sanity check, the environment is very quickly solved and in all cases the safety-aware algorithms quickly converge to the optimal reward of roughly $-22.0$, although PPO-Lag and CPO exhibit slightly slower convergence. For the bridge crossing environment both QL-Shield and QL-Cost are able to reliably find the path across the bridge, notice that this is a hard exploration challenge, and without penalties QL is unable to find the path across the bridge, both PPO-Lag and CPO also struggle with exploration.

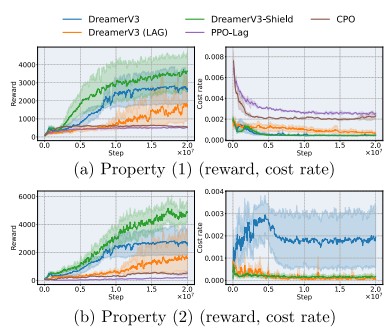

Figure 5: Learning curves for Seaquest.

For property (1) in the $9 \times 9$ gridworld, QL-shield is slightly more reliable than QL-Cost, as it converges to the shortest safe route more quickly, QL finds the shortest route very quickly, however this route is unsafe. For property (2) QL-Shield converges much more quickly than QL-Cost, this is likely because QL-Cost tries to find an overly conservative route that avoids any 'bomb' states, when in actuality it is allowed to step on 'bomb' states as long as it 'disarms' them. Note that PPO-Lag and CPO seem to do much better than for property (2) compared to property (1), as the safety criteria is not as strict.

For property (1) in the $15 \times 15$ gridworld QL-Shield and QL-Cost have a similar performance in terms of safety and reward, although QL-Shield is quite noisy, which suggests additional tuning of the step-wise safety rate $\varepsilon_t$ and $m$ could be useful. For property (2) QL-Shield converges quickly to a stable policy in contrast to QL-Cost, again this is likely because the QL-Cost is overly conservative tries to completely avoid bomb states. For property (3) QL-Shield does much better in terms of safety and QL-Cost doesn't appear to converge to a stable policy. Property (3) requires more effective exploration to find both the 'coloured' and 'medic' states, QL-Cost likely struggles to balance these two objectives with just one policy. Notice that PPO-Lag and CPO struggle here for all the properties as the problem requires much more effective exploration. For additional results see Appendix D.

For both property (1) and (2) in the Atari Seaquest environment our approach clearly outperforms the baselines in terms of reward and does well across the board in terms of safety performance. DreamerV3 (LAG) slightly outperforms our approach in terms of safety performance for property (2), however this is at the cost of much worse task performance (reward). Perhaps by using a stricter step-wise safety parameter $\varepsilon_t$ we could bring DreamerV3-Shield in line with DreamerV3 (LAG) for this property. PPO-Lag and CPO appear to do rather poorly in comparison, highlighting the poor sample complexity of model-free algorithms and demonstrating the difficulty with tuning the cost threshold $C$ and initial Lagrange multiplier $\lambda_{init}$.

## 5 RELATED WORK

**Safety Paradigms in Reinforcement Learning.** The most common paradigm is constrained MDPs (CMDP) for which, several constrained optimization algorithms have been developed, most are gradient-based methods built upon Lagrange relaxations of the constrained problem (Chow et al., 2018; Liang et al., 2018; Tessler et al., 2018; Ray et al., 2019) or projection-based local policy search (Achiam et al., 2017; Yang et al., 2020). Model-based approaches to CMDP (As et al., 2022; Huang et al., 2023; Thomas et al., 2021; Berkenkamp et al., 2017) have also gathered recent interest as they enjoy better sample complexity than their model-free counterparts (Janner et al., 2019).

Linear Temporal Logic (LTL) constraints (Voloshin et al., 2022; Hasanbeig et al., 2018; 2020a;b; De Giacomo et al., 2020; Cai et al., 2021) for RL have been developed as an alternative to CMDPs to specify stricter and more expressive constraints. The LTL formula is typically treated as the entire task specification, although some works have aimed to separate LTL satisfaction and reward into two distinct objectives (Voloshin et al., 2022; 2023; Shah et al., 2024). The typical procedure in this setting is to identify end components of the MDP that satisfy the LTL constraint and construct a corresponding reward function such that the optimal policy satisfies the LTL constraint with maximal probability. Formal PAC-style guarantees have been developed for this setting (Fu & Topcu, 2014; Wolff et al., 2012; Voloshin et al., 2022; Hasanbeig et al., 2018) although they often rely on non-trivial assumptions (e.g. access to a perfect generative model).

More rigorous safety-guarantees can be obtained by using *shielding* (Alshiekh et al., 2018), *control barrier functions* (CBF) (Ames et al., 2019), and *model predictive safety certification* (MPSC) (Wabersich & Zeilinger, 2018; 2021). To achieve zero-violation training, these methods typically assume that the dynamics of the system are known and thus they are typically restricted to low-dimensional systems. Recent works have aimed to scale the concept of shielding to more general settings, relaxing the prerequisite assumptions for shielding, by either only assuming access to a perfect generative model for planning (Giacobbe et al., 2021), or learning a world model from scratch (Goodall & Belardinelli, 2023; He et al., 2022; Xiao et al., 2023). Notable works that can be viewed as shielding include, MASE (Wachi et al., 2018) – a safe exploration algorithm with access to an 'emergency reset button', and Recovery-RL (Thananjeyan et al., 2021). A simple form of shielding with LTL specifications has also been considered (Mitta et al., 2024), although this approach makes use of informative priors over the transition dynamics. Shielding approaches most similar in spirit to our approach, include (Jansen et al., 2020; Könighofer et al., 2021; 2023), these approaches also consider finite-horizon satisfaction probabilities, although they assume a priori access to the safety dynamics and cannot provide episodic guarantees in the same way that we can.

**Learning Over Regular Structures.** RL and regular properties have been studied in conjunction before, perhaps most famously as 'Reward Machines' (Icarte et al., 2018; 2022) – a type of finite state automaton that specifies a different reward function at each automaton state, however reward machines do not explicitly deal with safety. In addition, regular decision processes (RDP) (Brafman et al., 2019) are a specific class non-Markov DPs (Bacchus et al., 1996) that have also been studied in several works (Brafman et al., 2019; Ronca & De Giacomo, 2021; Majeed et al., 2018; Toro Icarte et al., 2019; Cipollone et al., 2024). Most of these works are theoretical and slightly out-of-scope for this paper, as RDPs capture both non-Markov rewards and transition probabilities.

## 6 CONCLUSION

The separation of reward and safety objectives into two distinct policies has been demonstrated as an effective strategy towards safety-aware decision making (Goodall & Belardinelli, 2023; Jansen et al., 2018; Thananjeyan et al., 2021; Alshiekh et al., 2018), in many cases the safety objective is simpler and can be more quickly learnt (Jansen et al., 2018). In this paper we have demonstrated that this is an effective framework for dealing with regular safety properties, an important class of temporal properties where the corresponding cost function is non-Markov. We detail two possible instantiations of our framework for the tabular and deep RL environments, and we provide a thorough experimental evaluation including a comparison to CMDP-based approaches. Beyond our empirical results we provide safety guarantees in the tabular setting, that hold under reasonable assumptions. Future work includes, further investigation into the scenarios where it is appropriate and beneficial to leverage shielding as an approach to safe RL.

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

# A  ALGORITHMS

## A.1  QL-SHIELD

---

**Algorithm 1** QL-Shield (Regular Safety Property)

---

**Input:** DFA $\mathcal{D} = (\mathcal{Q}, \Sigma, \Delta, \mathcal{Q}_0, \mathcal{F})$, labelling function $L$, model checking parameters $(\varepsilon_t, \varepsilon', \delta', N)$, temperature $\tau > 0$, cost coefficient $c > 0$ and fixed episode length $T$

**Initialize:** (Q-table) $\hat{Q}_r(s, a) \leftarrow 0 \, \forall s \in \mathcal{S}, a \in \mathcal{A}$

**Initialize:** (Q-table) $\hat{Q}_b(s, q, a) \leftarrow 0 \, \forall s \in \mathcal{S}, q \in \mathcal{Q}, a \in \mathcal{A}$

**Initialize:** (Transition probabilities) $\widehat{\mathcal{P}} = \boldsymbol{I}$ (identity)

    **for each episode do**

        Observe $s_0$, $L(s_0)$ and $q_0 \leftarrow \Delta(\mathcal{Q}_0, L(s_0))$

        **for** $t = 0, \ldots, T$ **do**

            *// Sample an action from the 'task policy' and override if necessary*

            Sample action $a$ with the Boltzmann policy derived from $\hat{Q}_r(s_t, \cdot)$ and temp. $\tau$.

            $override \leftarrow \texttt{Shield}(\varepsilon_t, \varepsilon', \delta', N, \langle s_t, q_t \rangle, a, \pi_b, L, \mathcal{D}, \widehat{\mathcal{P}}, type = \texttt{statistical})$

            $a_t \leftarrow \arg\max_a Q_b(s_t, a)$ **if** *override* **else** $a_t \leftarrow a$

            Play action $a_t$ and observe $s_{t+1}$, $L(s_{t+1})$ and $r_t$.

            *// Update the 'task policy' and empirical probabilities*

            Update $\hat{Q}_r(s_t, a_t)$ with experience $(s_t, a_t, r_t, s_{t+1})$, see Eq. 3,

            Update $\widehat{\mathcal{P}}$ with experience $(s_t, a_t, s_{t+1})$, see Section 3.3.

            *// Counterfactual experiences (Icarte et al., 2022)*

            *// Generate synthetic data by simulating all automaton transitions*

            **for** $q \in \mathcal{Q}$ **do**

                Compute $q' \leftarrow \Delta(q, L(s_{t+1}))$

                Compute cost $c' \leftarrow c \cdot \mathbb{1}[q' \in \mathcal{F}]$

                *// Q-learning step*

                Update $\hat{Q}_b(s_t, q, a_t)$ with experience $(\langle s_t, q \rangle, a_t, \langle s_{t+1}, q' \rangle, c')$, see Eq. 4

            Compute $q_{t+1} \leftarrow \Delta(q_t, L(s_{t+1}))$ and continue

---

## A.2  MODEL CHECKING

---

**Algorithm 2** `Shield` (*type* = `statistical`)

---

**Input:** model checking parameters $(\varepsilon_t, \varepsilon' \, \delta', N)$, state $\langle s, q \rangle$, action $a$, 'backup policy' $\pi$, labelling function $L$, DFA $\mathcal{D} = (\mathcal{Q}, \Sigma, \Delta, \mathcal{Q}_0, \mathcal{F})$ and (approximate) transition probabilities $\mathcal{P}$.

    Choose $m \geq 2/(\varepsilon'^2) \log(2/\delta')$

    **for** $i = 1, \ldots, m$ **do**

        Set $s_0 \leftarrow s$, $q_0 \leftarrow q$ and $a_0 \leftarrow a$

        *// Sample a path through the model*

        **for** $j = 1, \ldots, N$ **do**

            Sample next state $s_j \sim \mathcal{P}(\cdot \mid s_{j-1}, a_{j-1})$,

            Compute $q_j \leftarrow \Delta(q_{j-1}, L(s_j))$,

            Sample action $a_j \sim \pi(\cdot \mid \langle s_j, q_j \rangle)$

        *// Check if the path is accepting*

        Let $X_i \leftarrow \mathbb{1}[q_H \in \mathcal{F}]$

    *// Compute the probability estimate*

    Let $\bar{X} \leftarrow \frac{1}{m} \sum_{i=1}^{m} X_i$

    *// If $\bar{X}$ is below the step-wise threshold we don't need to override*

    **return** *False* **if** $\bar{X} < \varepsilon_t - \varepsilon'$ **else return** *True*

---

---

**Algorithm 3** `Shield` (*type* = `exact`)

---

**Input:** model checking parameters $(\varepsilon_t, \varepsilon', \delta' = 0\ N)$, state $\langle s, q\rangle$, action $a$, 'backup policy' $\pi$, labelling function $L$, DFA $\mathcal{D} = (\mathcal{Q}, \Sigma, \Delta, \mathcal{Q}_0, \mathcal{F})$ and (approximate) transition probabilities $\mathcal{P}$.

    Compute the product MC: $\mathcal{M}_\pi \otimes \mathcal{D} = (\mathcal{S} \times \mathcal{Q}, \mathcal{P}', \mathcal{P}'_0, \{accept\}, L')$.

    Compute the probability matrix: $\boldsymbol{P} \leftarrow (\mathcal{P}'(s, t))_{s, t \notin accept}$

    Compute the probability vector: $\boldsymbol{p} \leftarrow (\mathcal{P}'(s, accept))_{s \notin accept}$

    Compute the conditional action matrix: $\boldsymbol{P}^{(a)} \leftarrow (\mathcal{P}(s, t)')_{s, t \notin accept}$

    $(\boldsymbol{P}^{(a)})_{\langle s, q\rangle} \leftarrow (\mathcal{P})_{\langle s, q\rangle, a} \cdot \pi(a \mid \langle s, q\rangle)$

    Compute the conditional action vector: $\boldsymbol{p}^{(a)} \leftarrow (\mathcal{P}'(s, accept))_{s \notin accept}$

    $(\boldsymbol{p}^{(a)})_{\langle s, q\rangle} \leftarrow (\mathcal{P})_{\langle s, q\rangle, a} \cdot \pi(a \mid \langle s, q\rangle)$

    *// Iterate over the model checking horizon*

    Initialize zero vector $\mathbf{x}^{(0)} \leftarrow \mathbf{0}$ with size $|\mathcal{S}| \times |\mathcal{Q}|$

    **for** $i = 1, \dots, N - 1$ **do**

        Compute $\mathbf{x}^{(i)} = \boldsymbol{P}\mathbf{x}^{(i-1)} + \boldsymbol{p}$

    *// Final update with the conditional action*

    Compute $\mathbf{x}^{(N)} = \boldsymbol{P}^{(a)}\mathbf{x}^{(N-1)} + \boldsymbol{p}^{(a)}$

    *// Get the corresponding probability*

    Let $X \leftarrow \mathbf{x}^{(N)}_{\langle s, q\rangle}$

    *// If $X$ is below the step-wise threshold we don't need to override*

    **return** *False* **if** $X < \varepsilon_t - \varepsilon'$ **else return** *True*

---

### A.3 DREAMERV3-SHIELD

---

**Algorithm 4** DreamerV3-Shielding (Regular Safety Property)

---

**Input:** DFA $\mathcal{D} = (\mathcal{Q}, \Sigma, \Delta, \mathcal{Q}_0, \mathcal{F})$, labelling function $L$, model checking parameters $(\varepsilon_t, \varepsilon', m, N)$, cost coefficient $c > 0$ and fixed episode length $T$, roll-out horizon $H$.

**Initialize:** replay buffer $D$, DreamerV3 parameters $\theta$, 'task policy' $\pi_r$ and 'backup policy' $\pi_b$.

    **for** each episode **do**

        Observe $o_0$, $L(s_0)$ and $q_0 \leftarrow \Delta(\mathcal{Q}_0, L(s_0))$

        **for** t = 1, …, T **do**

            *// Shielding with the latent world model*

            Sample action $a \sim \pi_r$ from the 'task policy'.

            Sample $m$ sequences $\langle\{\hat{o}_{t':t'+N}, \hat{r}_{t':t'+N}, \hat{c}_{t':t'+N}\}\rangle_{i=0}^{m} \sim p_\theta$ with $\pi_b$ and $a$.

            *// Compute the probability estimate*

            $\bar{X} \leftarrow \frac{1}{m}\sum_{i=0}^{m} clip\left(\sum_{t'}^{t'+N} c_{t'}, 0.0, 1.0\right)$

            *override* $\leftarrow$ *False* **if** $\bar{X} < \varepsilon_t - \varepsilon'$ **else** *True*

            $a_t \sim \pi_b$ **if** *override* **else** $a_t \leftarrow a$

            Play action $a_t$ and observe $o_{t+1}$, $L(s_{t+1})$ and $r_t$

            Compute $q_{t+1} \leftarrow \Delta(q_t, L(s_{t+1}))$,

            Compute cost $c \cdot c_t \leftarrow \mathbb{1}[q_{t+1} \in \mathcal{F}]$

            Append $(o_t, a_t, r_t, c_t, o_{t+1})$ to the replay buffer $D$

            **if** update **then**

                *// World model learning*

                Sample a batch $B$ of transition sequences $\{(o_{t'}, a_{t'}, r_{t'}, c_{t'}, o_{t'+1})\} \sim \mathcal{D}$.

                Update DreamerV3 parameters $\theta$ with maximum likelihood (Hafner et al., 2023).

                *// Task policy optimization*

                Sample sequences $\{\hat{o}_{t':t'+H}, \hat{r}_{t':t'+H}, \hat{c}_{t':t'+H}\} \sim p_\theta$ with the 'task policy' $\pi_r$

                Update the 'task policy' $\pi_r$ with RL (to maximize reward).

                Update the corresponding value critics with maximum likelihood

                *// Backup policy optimization*

                Sample sequences $\{\hat{o}_{t':t'+H}, \hat{r}_{t':t'+H}, \hat{c}_{t':t'+H}\} \sim p_\theta$ with the 'backup policy' $\pi_b$

                Update the 'backup policy' $\pi_b$ with RL (to minimize cost)

                Update the corresponding value critics with maximum likelihood

---

# B  PROOFS

## B.1  PROOF OF PROPOSITION 3.5

**Proposition 3.5 (restated).** *Let $\varepsilon' > 0$, $\delta' > 0$, $\langle s, q \rangle \in \mathcal{S} \times \mathcal{Q}$ and $N \geq 1$ be given. By sampling $m \geq \frac{1}{2\varepsilon'^2} \log \left( \frac{2}{\delta'} \right)$ many paths with $\mathcal{P}$, we can obtain an $\varepsilon'$-approximate estimate for the probability $\mathrm{Pr}_{\pi_b}^a (\langle s, q \rangle \models \Diamond^{\leq N} accept)$ with probability at least $1 - \delta'$.*

*Proof.* In words, we estimate $\mathrm{Pr}_{\pi_b}^a (\langle s, q \rangle \models \Diamond^{\leq N} accept)$ by sampling $m$ paths with $\mathcal{P}$, first using the action $a$ to resolve the non-determinism of the MDP and then using the fixed policy $\pi_b$ thereafter. We can simply label each path as satisfying or not and return the proportion of satisfying traces as our estimate for $\mathrm{Pr}_{\pi_b}^a (\langle s, q \rangle \models \Diamond^{\leq N} accept)$.

We proceed as follows, let $\rho_1, \ldots \rho_m$ be a sequence of paths sampled from the MDP as described above and let $trace(\rho_1), \ldots trace(\rho_m)$ be the corresponding traces. Furthermore, let $X_1, \ldots, X_m$ be indicator r.v.s such that,

$$X_i = \begin{cases} 1 & \text{if } trace(\rho_i) \models \Diamond^{\leq N} accept, \\ 0 & \text{otherwise} \end{cases} \tag{7}$$

Note that $trace(\rho_1) \models \Diamond^{\leq N} accept$ can be easily checked in time $O(N)$. Now let,

$$\bar{X} = \frac{1}{m} \sum_{i=1}^{m} X_i \text{ where } \mathbb{E}[\bar{X}] = \mathrm{Pr}_{\pi_b}^a (\langle s, q \rangle \models \Diamond^{\leq N} accept) \tag{8}$$

then by Hoeffding's inequality (Hoeffding, 1963),

$$\mathbb{P}\left[ |\bar{X} - \mathbb{E}[\bar{X}]| \geq \varepsilon' \right] \leq 2 \exp\left( -2m\varepsilon'^2 \right) \tag{9}$$

Bounding the RHS from above by $\delta'$ and rearranging gives the desired result. $\qquad\square$

## B.2  PROOF OF PROPOSITION 3.6

We start by introducing the following lemma.

**Lemma B.1** (Error amplification for trace distributions). *Let $\widehat{\mathcal{P}} \approx \mathcal{P}$ be such that,*

$$D_{TV}\left( \mathcal{P}(\cdot \mid s, a), \widehat{\mathcal{P}}(\cdot \mid s, a) \right) \leq \alpha \; \forall (s, a) \in S \times \mathcal{A} \tag{10}$$

*Let the start state $s_0 \in \mathcal{S}$ be given, let the policy $\pi$ be given and let $\mathcal{P}_\pi^t(\cdot)$ and $\widehat{\mathcal{P}}_\pi^t(\cdot)$ denote the path distribution (at time $t$) for the two Markov chain transition probabilities $\mathcal{P}_\pi$ and $\widehat{\mathcal{P}}_\pi$ respectively. Then the total variation distance between the two path distributions (at time $t$) is bounded as follows,*

$$D_{TV}\left( \mathcal{P}_\pi^t(\cdot), \widehat{\mathcal{P}}_\pi^t(\cdot) \right) \leq \alpha t \; \forall t \tag{11}$$

*Proof.* We will prove this fact by doing an induction on $t$. We recall that $\mathcal{P}_\pi^t(\cdot)$ and $\widehat{\mathcal{P}}_\pi^t(\cdot)$ denote the path distribution (at time $t$) for the two transition probabilities $\mathcal{P}_\pi$ and $\widehat{\mathcal{P}}_\pi$ respectively. Formally we define them as follows,

$$\mathcal{P}_\pi^t(\rho) = \mathrm{Pr}(s_0, \ldots, s_t \preceq \rho \mid s_0 = s, \mathcal{P}_\pi) \tag{12}$$

$$\widehat{\mathcal{P}}_\pi^t(\rho) = \mathrm{Pr}(s_0, \ldots, s_t \preceq \rho \mid s_0 = s, \widehat{\mathcal{P}}_\pi) \tag{13}$$

These probabilities read as follows, 'the probability of the sequence $s_0, \ldots, s_t \preceq \rho$ at time $t$', or similarly 'the probability that the sequence $s_0, \ldots, s_t$ is a prefix of $\rho$ at time $t$' Since the start state $s_0 \in \mathcal{S}$ is given we note that,

$$\mathcal{P}_\pi^0(\cdot) = \widehat{\mathcal{P}}_\pi^0(\cdot) \tag{14}$$

Before we continue with the induction on $t$ we make the following observation, for any path $\rho \in \mathcal{S}^\omega$ we have by the triangle inequality,

$$\left| \mathcal{P}_\pi^t(\rho) - \widehat{\mathcal{P}}_\pi^t(\rho) \right| = \left| \mathcal{P}_\pi(s_t \mid s_{t-1}) \mathcal{P}_\pi^{t-1}(\rho) - \widehat{\mathcal{P}}_\pi(s_t \mid s_{t-1}) \widehat{\mathcal{P}}_\pi^{t-1}(\rho) \right| \tag{15}$$

$$\leq \mathcal{P}_\pi^{t-1}(\rho) \left| \mathcal{P}_\pi(s_t \mid s_{t-1}) - \widehat{\mathcal{P}}_\pi(s_t \mid s_{t-1}) \right| + \widehat{\mathcal{P}}_\pi(s_t \mid s_{t-1}) \left| \mathcal{P}_\pi^{t-1}(\rho) - \widehat{\mathcal{P}}_\pi^{t-1}(\rho) \right| \tag{16}$$

Now we continue with the induction on $t$,

$$2 D_{TV}(\mathcal{P}_\pi^t(\cdot), \widehat{\mathcal{P}}_\pi^t(\cdot)) = \sum_{\rho \in \mathcal{S}^\omega} \left| \mathcal{P}_\pi^t(\rho) - \widehat{\mathcal{P}}_\pi^t(\rho) \right| \tag{17}$$

$$\leq \sum_{\rho \in \mathcal{S}^\omega} \mathcal{P}_\pi^{t-1}(\rho) \left| \mathcal{P}_\pi(s_t \mid s_{t-1}) - \widehat{\mathcal{P}}_\pi(s_t \mid s_{t-1}) \right|$$
$$+ \sum_{\rho \in \mathcal{S}^\omega} \widehat{\mathcal{P}}_\pi(s_t \mid s_{t-1}) \left| \mathcal{P}_\pi^{t-1}(\rho) - \widehat{\mathcal{P}}_\pi^{t-1}(\rho) \right| \tag{18}$$

$$\leq \sum_{\rho \in \mathcal{S}^\omega} \mathcal{P}_\pi^{t-1}(\rho) \cdot (2\alpha) + \sum_{\rho \in \mathcal{S}^\omega} \left| \mathcal{P}_\pi^{t-1}(\rho) - \widehat{\mathcal{P}}_\pi^{t-1}(\rho) \right| \tag{19}$$

$$= 2\alpha + 2 D_{TV}(\mathcal{P}_\pi^{t-1}(\cdot), \widehat{\mathcal{P}}_\pi^{t-1}(\cdot)) \tag{20}$$

$$\leq 2\alpha t \tag{21}$$

The final result is obtained by an induction on $t$ where the base case comes from $\mathcal{P}_\pi^0(\cdot) = \widehat{\mathcal{P}}_\pi^0(\cdot)$. $\quad\square$

**Proposition 3.6 (restated).** *Let $\varepsilon' > 0$, $\delta' > 0$, $s \in \mathcal{S}$ and $N \geq 1$ be given. Suppose that for all $(s, a) \in \mathcal{S} \times \mathcal{A}$, our empirical estimate $\widehat{\mathcal{P}}$ is such that,*

$$D_{TV}\left( \mathcal{P}(\cdot \mid s, a), \widehat{\mathcal{P}}(\cdot \mid s, a) \right) \leq \varepsilon'/N \tag{22}$$

*where $D_{TV}$ denotes the total variation (TV) distance, then,*

*(1) We can obtain an $\varepsilon'$-approximate estimate for $\mathrm{Pr}_{\pi_b}^a(\langle s, q \rangle \models \Diamond^{\leq N} accept)$ with probability 1 by exact model checking with the transition probabilities of $\widehat{\mathcal{P}}$ in time $\mathcal{O}(poly(size(\mathcal{M}_\pi \otimes \mathcal{D})) \cdot N)$.*

*(2) We can obtain an $\varepsilon'$-approximate estimate for $\mathrm{Pr}_{\pi_b}^a(\langle s, q \rangle \models \Diamond^{\leq N} accept)$ with probability at least $1 - \delta'$, by sampling $m \geq \frac{2}{\varepsilon'^2} \log\left( \frac{2}{\delta'} \right)$ many paths with the 'approximate model' $\widehat{\mathcal{P}}$.*

*Proof.* We start by proving statement (1) and then statement (2) will follow quickly. First let $\mathrm{Pr}_{\pi_b}^a(\langle s, q \rangle \models \Diamond^{\leq N} accept)$ and $\widehat{\mathrm{Pr}_{\pi_b}^a}(\langle s, q \rangle \models \Diamond^{\leq N} accept)$ denote the conditional action probabilities for the two transition probabilities $\mathcal{P}$ and $\widehat{\mathcal{P}}$ respectively. We also let $g(\cdot)$ and $\widehat{g}(\cdot)$ denote the average trace distribution (over the next $N$ timesteps) for the two transition probabilities $\mathcal{P}$ and $\widehat{\mathcal{P}}$ respectively, where,

$$g(\rho) = \frac{1}{N} \sum_{t=1}^{N} \mathcal{P}_{\pi_b}^t(\rho) \tag{23}$$

$$\widehat{g}(\rho) = \frac{1}{N} \sum_{t=1}^{N} \widehat{\mathcal{P}}_{\pi_b}^t(\rho) \tag{24}$$

Abusing notation slightly (by dropping $a$), we note that in both instances the action $a$ is first used to resolve the non-determinism of the MDP and $\pi_b$ thereafter. Before we continue with the proof of (1) we make the following observations,

- $\displaystyle \max_{\langle s, q \rangle} \left| \mathrm{Pr}_{\pi_b}^a(\langle s, q \rangle \models \Diamond^{\leq N} accept) - \widehat{\mathrm{Pr}_{\pi_b}^a}(\langle s, q \rangle \models \Diamond^{\leq N} accept) \right| \leq 1$

- Let $f(x) : x \in \mathcal{X} \to [0, 1]$ be a real-valued function. Let $\mathcal{P}_1(\cdot)$ and $\mathcal{P}_2(\cdot)$ be probability distributions over the space $\mathcal{X}$, then.
$$\left| \mathbb{E}_{x \sim \mathcal{P}_1(\cdot)}[f(x)] - \mathbb{E}_{x \sim \mathcal{P}_2(\cdot)}[f(x)] \right| \leq D_{TV}(\mathcal{P}_1(\cdot), \mathcal{P}_2(\cdot))$$

We continue by showing the following,

$$\left| \Pr_{\pi_b}^a(\langle s, q\rangle \models \Diamond^{\leq N} accept) - \widehat{\Pr}_{\pi_b}^a(\langle s, q\rangle \models \Diamond^{\leq N} accept) \right| \tag{25}$$

$$= \left| \mathbb{E}_{\rho \sim g} \left[ \mathbb{1} \left[ \langle s, q\rangle \models \Diamond^{\leq N} accept \right] \right] - \mathbb{E}_{\rho \sim \widehat{g}} \left[ \mathbb{1} \left[ \langle s, q\rangle \models \Diamond^{\leq N} accept \right] \right] \right| \tag{26}$$

$$\leq D_{TV}\left( g(\cdot), \widehat{g}(\cdot) \right) \tag{27}$$

$$= \frac{1}{2} \sum_{\rho \in \mathcal{S}^\omega} |g(\rho) - \widehat{g}(\rho)| \tag{28}$$

$$= \frac{1}{2N} \sum_{\rho \in \mathcal{S}^\omega} \left| \sum_{t=1}^{N} \mathcal{P}_\pi^t(\rho) - \widehat{\mathcal{P}}_\pi^t(\rho) \right| \tag{29}$$

$$\leq \frac{1}{2N} \sum_{t=1}^{N} \left| \sum_{\rho \in \mathcal{S}^\omega} \mathcal{P}_\pi^t(\rho) - \widehat{\mathcal{P}}_\pi^t(\rho) \right| \tag{30}$$

$$\leq \frac{1}{2N} \sum_{t=1}^{H} N(\varepsilon'/N) \tag{31}$$

$$= \varepsilon'/2 \tag{32}$$

$$\tag{33}$$

The first inequality (27) comes from our earlier observations. The second inequality (30) is straightforward and the final inequality (31) is obtained by applying Lemma B.1 and our initial assumption in (22). We note that this result is closely related to the *simulation lemma* (Kearns & Singh, 2002), which has been proved many times for several different settings (Kakade et al., 2003; Abbeel & Ng, 2005; Brunskill et al., 2009; Rajeswaran et al., 2020).

This concludes the proof of statement (1), since we have shown that $\widehat{\Pr}_{\pi_b}^a(\langle s, q\rangle \models \Diamond^{\leq N} accept)$ is an $\varepsilon'/2$-approximate estimate of $\Pr_{\pi_b}^a(\langle s, q\rangle \models \Diamond^{\leq N} accept)$, under the our initial assumption in (22).

The proof of statement (2) follows quickly. We have established that,

$$\left| \Pr_{\pi_b}^a(\langle s, q\rangle \models \Diamond^{\leq N} accept) - \widehat{\Pr}_{\pi_b}^a(\langle s, q\rangle \models \Diamond^{\leq N} accept) \right| \leq \varepsilon'/2 \tag{34}$$

It remains to obtain an $\varepsilon'/2$-approximate estimate of $\widehat{\Pr}_{\pi_b}^a(\langle s, q\rangle \models \Diamond^{\leq N} accept)$. By using the same reasoning as in the proof of Proposition 3.5. We can obtain an $\varepsilon'/2$-approximate estimate of $\widehat{\Pr}_{\pi_b}^a(\langle s, q\rangle \models \Diamond^{\leq N} accept)$ by sampling $m$ paths, $\rho_1, \ldots \rho_m$, from the approximate dynamics model $\widehat{\mathcal{P}}$. Then provided,

$$m \geq \frac{2}{\varepsilon'^2} \log\left( \frac{2}{\delta'} \right) \tag{35}$$

with probability $1 - \delta'$ we can obtain $\varepsilon'/2$-approximate estimate of $\widehat{\Pr}_{\pi_b}^a(\langle s, q\rangle \models \Diamond^{\leq N} accept)$ and by extension an $\varepsilon'$-approximate estimate of $\Pr_{\pi_b}^a(\langle s, q\rangle \models \Diamond^{\leq N} accept)$. This concludes the proof. $\qquad\square$

### B.3 PROOF OF THEOREM 3.11

**Theorem 3.11 (restated).** *Under Assumption 3.9 and 3.10, and provided that every state action pair $(s, a) \in \mathcal{S} \times \mathcal{A}$ has been visited at least $\mathcal{O}\left( \frac{N^2 |\mathcal{S}|}{\varepsilon'^2} \log\left( \frac{|\mathcal{A}||\mathcal{S}|}{\delta'} \right) \right)$ times. Then the 'shielded policy' $\pi_{sh}$ provides a step-wise safety guarantee of $\varepsilon_t$ and with a step-wise failure probability of $\delta_t = 2\delta'$.*

*Proof.* We split the proof up into three parts **(1)**, **(2)**, **(3)**.

**(1)** We first show that the following holds with probability at least $1 - \delta'$,

$$D_{TV}\left( \mathcal{P}(\cdot \mid s, a), \widehat{\mathcal{P}}(\cdot \mid s, a) \right) \leq \varepsilon'/N \tag{36}$$

when every state action pair $(s, a) \in \mathcal{S} \times \mathcal{A}$ has been visited at least,

$$\mathcal{O}\left(\frac{N^2|\mathcal{S}|}{\varepsilon'^2} \log\left(\frac{|\mathcal{A}||\mathcal{S}|}{\delta'}\right)\right)$$

times. First we let $\#(s, a)$ denote the total number of times that $(s, a)$ has been observed, similarly we let $\#(s', s, a)$ denote the total number of times that $(s', s, a)$ has been observed. The maximum likelihood estimate for the unknown probability $\mathcal{P}(s' \mid, s, a)$ is $\widehat{\mathcal{P}}(s' \mid s, a) = \#(s', s, a)/\#(s, a)$. Let us fix some $(s, a) \in \mathcal{S} \times \mathcal{A}$, leveraging the well-established inequality for the L1 deviation from the empirical distribution (Weissman et al., 2003), we can bound the TV distance between $\mathcal{P}(\cdot \mid, s, a)$ and $\widehat{\mathcal{P}}(\cdot \mid s, a)$ as follows,

$$\mathbb{P}\left[\left\|\mathcal{P}(\cdot \mid s, a) - \widehat{\mathcal{P}}(\cdot \mid s, a)\right\|_1 \geq \frac{\varepsilon'}{N}\right] \leq (2^{|S|} - 2) \exp\left(\frac{-m\varepsilon'^2}{2N^2}\right) \tag{37}$$

Bounding the RHS from above by $\delta'/(|A||S|)$ and rearranging gives the following lower bound for $m$,

$$m \geq \frac{2N^2}{\varepsilon'^2} \log\left(\frac{|S||A|(2^{|S|} - 2)}{\delta'}\right) = \mathcal{O}\left(\frac{N^2|S|}{\varepsilon'^2} \log\left(\frac{|S||A|}{\delta'}\right)\right) \tag{38}$$

Taking a union bound over all $(s, a) \in \mathcal{S} \times \mathcal{A}$, then for all state action pairs $(s, a) \in \mathcal{S} \times \mathcal{A}$ we have the following with probability at least $1 - \delta'$.

$$D_{TV}\left(\mathcal{P}(\cdot \mid s, a), \widehat{\mathcal{P}}(\cdot \mid, s, a)\right) = \frac{1}{2} \sum_{s' \in S} \left|\mathcal{P}(s' \mid s, a) - \widehat{\mathcal{P}}(s' \mid s, a)\right| \tag{39}$$

$$\leq \frac{1}{2}\left\|\mathcal{P}(\cdot \mid s, a) - \widehat{\mathcal{P}}(\cdot \mid s, a)\right\|_1 \tag{40}$$

$$\leq \varepsilon'/N \tag{41}$$

This completes the proof for part **(1)**.

**(2)** Now by using Assumption 3.9 and 3.10 we can reason about the safety of the system. Suppose firstly that we can exactly compute the conditional action probability $\Pr_{\pi_b}^a(\langle s, q \rangle \models \Diamond^{\leq N} accept)$ and without any failure probability – this corresponds to exact model checking with the transition probabilities $\mathcal{P}$.

Under Assumption 3.10 the initial state $\langle s_0, L(s_0) \rangle$ is contained in the *probabilistic safe set* $\mathcal{S}^{\pi_b}(\varepsilon_t)$ meaning that by following the 'backup policy' $\pi_b$ we can satisfy the safety property $P_{safe}$ for the entire episode length with probability at least $1 - \varepsilon_t$.

The 'shielded policy' $\pi_{sh}$ is constructed such that an action $a$ proposed by the 'task policy' $\pi_r$ is only permissible if $\Pr_{\pi_b}^a(\langle s, q \rangle \models \Diamond^{\leq N} accept) \leq \varepsilon_t$,

$$\pi_{sh}(\langle s, q \rangle, a) = \begin{cases} \pi_r(s, a) & \text{if } \Pr_{\pi_b}^a(\langle s, q \rangle \models \Diamond^{\leq N} accept) \leq \varepsilon_t \\ \pi_b(\langle s, q \rangle, a) & \text{otherwise} \end{cases} \tag{42}$$

Under Assumption 3.9 any permissible action $a$ proposed by the 'task policy' $\pi_r$ is 'safe' in the sense that $\langle s, q \rangle$ will be contained in the *probabilistic safe set* $\mathcal{S}^{\pi_b}(\varepsilon_t)$. The reasoning for this is straightforward proof by contradiction, assume $\Pr_{\pi_b}^a(\langle s, q \rangle \models \Diamond^{\leq N} accept) \leq \varepsilon_t$ and $\Pr_{\pi_b}^a(\langle s, q \rangle \models \Diamond^{\leq T} accept) > \varepsilon_t$ then by definition the action $a$ is irrecoverable and so by Assumption 3.9 we must have $\Pr_{\pi_b}^a(\langle s, q \rangle \models \Diamond^{\leq N^*} accept) > \varepsilon_t$, however since $N \geq N^*$ then certainly $\Pr_{\pi_b}^a(\langle s, q \rangle \models \Diamond^{\leq N} accept) > \Pr_{\pi_b}^a(\langle s, q \rangle \models \Diamond^{\leq N^*} accept) > \varepsilon_t$ which is a contradiction.

Thus if a permissible action $a$ proposed by the 'task policy' $\pi_r$ is committed in the environment then we know that the current state $\langle s, q \rangle$ is contained in the *probabilistic safe set* $\mathcal{S}^{\pi_b}(\varepsilon_t)$ and thus we have established the following invariant: 'we can always fall back on the backup policy for a step-wise safety guarantee of $\varepsilon_t$ regardless of the previous action'.

**(3)** We we make a similar argument for exact model checking with the empirical probabilities $\widehat{\mathcal{P}}$, where we can only obtain an $\varepsilon'$-approximate estimate of the conditional action probability $\Pr_{\pi_b}^a(\langle s, q \rangle \models \Diamond^{\leq N} accept)$. The key to this part of the proof is to only allow actions proposed by the 'task policy' $\pi_r$ we know for certain satisfy $\Pr_{\pi_b}^a(\langle s, q \rangle \models \Diamond^{\leq N} accept) \leq \varepsilon_t$.

In particular an action $a$ proposed by the 'task policy' $\pi_r$ is only permissible if our estimate for $\text{Pr}^a_{\pi_b}(\langle s, q \rangle \models \Diamond^{\leq N} accept)$ denoted $\widehat{\text{Pr}^a_{\pi_b}}(\langle s, q \rangle \models \Diamond^{\leq N} accept)$, is less than $\varepsilon_t - \varepsilon'$, this decision is reflected in both Algorithm 3 and 2 in Appendix A. If $\widehat{\text{Pr}^a_{\pi_b}}(\langle s, q \rangle \models \Diamond^{\leq N} accept) \leq \varepsilon_t - \varepsilon'$ then $\text{Pr}^a_{\pi_b}(\langle s, q \rangle \models \Diamond^{\leq N} accept) \leq \varepsilon_t$, the proof of this statement is a straightforward proof by contradiction, assume that $\widehat{\text{Pr}^a_{\pi_b}}(\langle s, q \rangle \models \Diamond^{\leq N} accept) \leq \varepsilon_t - \varepsilon'$ and $\text{Pr}^a_{\pi_b}(\langle s, q \rangle \models \Diamond^{\leq N} accept) > \varepsilon_t$, then we have $|\widehat{\text{Pr}^a_{\pi_b}}(\langle s, q \rangle \models \Diamond^{\leq N} accept) - \text{Pr}^a_{\pi_b}(\langle s, q \rangle \models \Diamond^{\leq N} accept)| > \varepsilon'$ which is a contradiction as we have established in Proposition 3.6 that $\widehat{\text{Pr}^a_{\pi_b}}(\langle s, q \rangle \models \Diamond^{\leq N} accept)$ is an $\varepsilon'$-approximate estimate of $\text{Pr}^a_{\pi_b}(\langle s, q \rangle \models \Diamond^{\leq N} accept)$ when (36) is satisfied.

**Putting it all together.** Part **(1)** of our proof establishes that with probability at least $1 - \delta'$ the total variation distance between $\mathcal{P}$ and $\widehat{\mathcal{P}}$ is upper bounded, see (36). Part **(3)** then establishes how we can use the $\varepsilon'$-approximate estimate of the conditional action probability $\text{Pr}^a_{\pi_b}(\langle s, q \rangle \models \Diamond^{\leq N} accept)$ to only let permissible actions be used by the 'shielded policy', this in conjunction with the invariant established in part **(2)** completes the proof for exact model checking. We finally need to deal with the failure probability associated with statistical model checking. In particular, at each timestep we fix a failure probability of $\delta'$, taking a union bound with part **(1)** of the proof gives us a step-wise failure probability of $\delta_t = 2\delta'$. The completes the proof. $\square$

## C  ENVIRONMENT DESCRIPTIONS

In this section we provide more complete descriptions of the environments considered in the main paper.

**Media streaming.** Inspired by Bura et al. (2022), The agent is tasked with managing a data-buffer, packets leave in the data-buffer according to a Bernoulli process with rate $\mu_{out}$, the agent has two action $\mathcal{A} = \{fast, slow\}$ which add new packets to the data-buffer according to a Bernoulli process with rates $\mu_{fast} = 0.9$ and $\mu_{slow} = 0.1$ respectively. The agent receives a negative reward of $-1.0$ for choosing the fast rate, the goal is to maximize reward during the fixed episode length $T = 40$, while ensuring the data-buffer is never empty. The safety property is a simple invariant property, $\square \neg empty$ (with PCTL-style notation). The number of automaton states is $|\mathcal{D}| = 2$, the safety threshold (for QL-Shield) is set to $\varepsilon_t = 0.001$ and the cost threshold for PPO-Lag and CPO is set to $C = 0.01$ respectively. The model checking horizon we use here is $N = 5$. For further hyperparameter details please refer to Appendix E.3.

**Bridge crossing.** Inspired by Hasanbeig et al. (2020a), the agent operates in a $20 \times 20$ 'slippery' gridworld where there is a $0.04$ chance that the agent's action is ignored and another action is uniformly sampled. From the green start state the goal is to reach the safe terminal yellow states, which provide a reward $+1$. The unsafe red states are also terminal (providing no reward). The safety property is a simple invariant $\square \neg red$(with PCTL-style notation). The number of automaton states is $|\mathcal{D}| = 2$, the safety threshold (for QL-Shield) is set to $\varepsilon_t = 0.05$ and the cost threshold for PPO-Lag and CPO is set to $C = 0.15$. The model checking horizon we use here is $N = 5$.

$9 \times 9$ **gridworld.** The agent operates in a $9 \times 9$ 'slippery' gridworld where there is a $0.1$ chance that the agent's action is ignored. From the start state $S$ the goal is to reach either the *blue*, *pink* or *yellow* states which are terminal and provide a reward of $+1$. For this environment we use the following two properties the first (1) is a simple invariant property $\square \neg B$, the second (2) is $\square((\neg B X B) \to (X B))$. In words, (1) specifies that the agent must avoid 'bomb' states ($B$), (2) specifies that the agent must 'disarm' 'bomb' states ($B$) by staying on them for at least 2 timesteps. For (1): $|\mathcal{D}| = 2$ and $\varepsilon_t = 0.01$, $C = 0.01$ and $N = 3$, for (2): $|\mathcal{D}| = 4$ and $\varepsilon_t = 0.12$, $C = 0.12$ and $N = 5$.

$15 \times 15$ **gridworld.** The agent operates in a $15 \times 15$ 'slippery' gridworld where there is a $0.1$ chance that the agent's action is ignored. The goal is to reach either the *blue*, *pink*, *yellow*, *red* or *green* states (providing a reward of $+1$) from any of the starting states ($S$). In this environment the goal states are no longer terminal and reaching a goal state transitions the agent to a new start state ($S$) sampled uniformly at random. The agent's goal is to collect as much reward in the fixed episode length of $T = 250$. For this environment we experiment with three properties. Property (1) and (2) are identical to the $9 \times 9$ gridworld (see above). The third property (3) specifies that if the agent reaches a 'bomb' state ($B$) then must reach and stay in a 'medic' state ($M$) for two timesteps, within

10 timesteps, with PCTL-style notation this is denoted as $\Box(B \to \Diamond^{\leq 10}\Box^{\leq 2}M)$. For (3) we have $|\mathcal{D}| = 22$ and $\varepsilon_t = 0.001$, $C = 0.01$ and $N = 13$.

**Seaquest.** Seaquest is an Atari 2600 games provided by the Arcade Learning Environment (ALE) (Machado et al., 2018). Seaquest utilizes the full Atari controller action space (18 distinct actions), which includes all possible combinations of FIRE, UP, DOWN, LEFT, RIGHT. The observations provided by the ALE are $210 \times 160 \times 3$ tensors which correspond to the RGB values of each pixel on the screen. The observations are scaled to $64 \times 64 \times 3$ pixel images with the RGB information intact. `frame_skip=4` (every 4 frames are skipped with the agent's action repeated) and `sticky_actions=0.25` (there is a 0.25 chance that the agent's previous action is repeated) are used, which are in line with the recommended settings for Atari games (Machado et al., 2018). In Seaquest the goal is to collect reward by rescuing divers and 'shooting' enemy sharks and submarines. In addition to collecting reward, the agent must manage its oxygen resources and avoid being hit by sharks and the enemy submarines which fire back. In our experiments we evaluated two regular safety properties, *"(1) ($\Box\neg surface \to \Box(surface \to diver)) \wedge (\Box\neg out\text{-}of\text{-}oxygen) \wedge (\Box\neg hit)$, and (2) $\Box diver \wedge \neg surface \to \Diamond^{\leq 30}surface$. The first property (1) is aligned closely with the goal – the agent must only surface with a diver, not run out of oxygen and not be hit by an enemy. The second property (2) states after the agent picks up a diver it must return to the surface within 30 timesteps, this property directly conflicts with the optimal policy."* For (1) the size of the DFA is $|\mathcal{D}| = 4$ and for (2) the size of the DFA is $|\mathcal{D}| = 30$. For both properties the safety threshold (for DreamerV3-Shield) is set to $\varepsilon_t = 0.01$ and the cost threshold (for DreamerV3 (Lag), PPO-Lag and CPO) is set to $C = 1.0$.

## C.1 Additional Environments

**Pacman** This environment is inspired by Voloshin et al. (2022), see Fig. 6. The agent (red triangle) operates in a Pacman world, with one ghost and one piece of food (yellow circle) to collect. A reward of $+1$ is obtained by picking up the food, once the food is picked up the food disappears and the agent must avoid the ghost for the remainder of the fixed episode length $T = 100$, this safety property is specified by the simple invariant property $\Box\neg ghost$ (with PCTL-style notation). The size of the DFA is $|\mathcal{D}| = 2$. Including all possible directions and locations for the agent and the ghost, and whether or not the food has been picked up yet, there are approximately

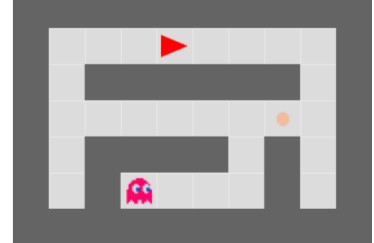

Figure 6: Pacman environment from (Voloshin et al., 2022).

8000 states. The agent can pick one of the following actions UP, DOWN, LEFT, RIGHT, STAY. Although, note, as in standard Pacman navigation, the the agent can only turn around (180 degrees) when they are facing a wall, limiting the navigational ability of the agent when compared to standard gridworld settings. The ghost has the same restrictions, however the ghost chases the agent (following the shortest path) with probability $0.4$ and chooses a random action uniformly from its available moves with probability $0.6$. The environment, is actually more challenging that you might imagine, due to the random nature of the ghost, it is difficult force the ghost to follow the agent at which point collecting the food becomes easy. For QL-Shield we use a step-wise safety-threshold of $\varepsilon_t = 0.01$ and model checking horizon $N = 10$. For this environment we provide the results in Appendix D.4.

# D Additional Results and Ablation Studies

In this section we conduct a set of ablation studies, in particular, we conduct experiments in the tabular gridworld environments, where in contrast to QL-Shield, we are given access to the transition probabilities $\mathcal{P}$ and an optimal safe 'backup policy' denoted $\pi_b^*$, which is constructed with value iteration before training of the 'task policy' $\pi_r$. We also use exact PCTL model checking to compute the conditional action probability $\Pr_{\pi_b^*}^a(\langle s, q \rangle \models \Diamond^{\leq N} accept)$ when shielding the 'task policy'. Since $\mathcal{P}$ and $\pi_b^*$ are fixed during learning, we can actually compute an action satisfaction set and verify that Assumption 3.9 and 3.10 do in fact hold. This gives us a step-wise safety guarantee of $\varepsilon_t$ at the start of training, which will be reflected in our experimental results.

We call this instantiation of our framework QL-Exact. The assumption of prior knowledge of $\mathcal{P}$ of course does not fit in to the general RL framework, however it is interesting to see how quickly QL-Shield (which is compatible with the typical RL framework) converges to the performance of QL-Exact. We note that for QL-Exact the 'task policy' $\pi_r$ is not 'pre-trained' and so the task performance of QL-Exact is not immediately optimal. We provide the results below; we plot the reward, the cost, the 'episodic' safety rate and the episode length where relevant.

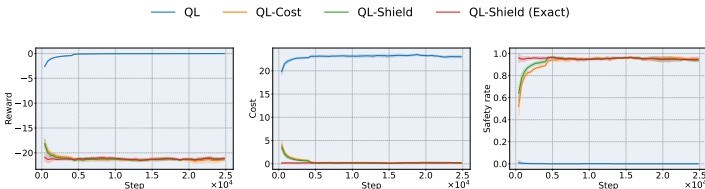

Figure 7: Ablation study with QL-Exact for Media Streaming.

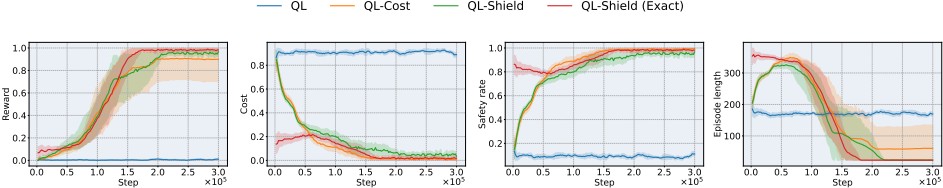

Figure 8: Ablation study with QL-Exact for Bridge Crossing.

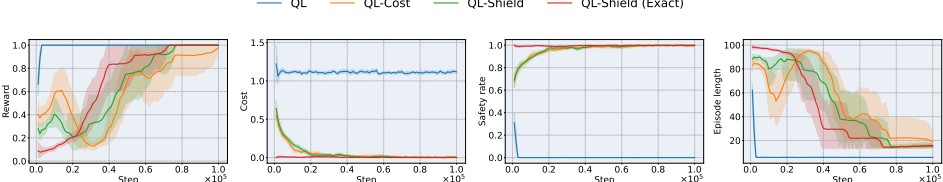

Figure 9: Ablation study with QL-Exact for $9 \times 9$ gridworld property (1) (reward, cost, episodic safety rate, episode length).

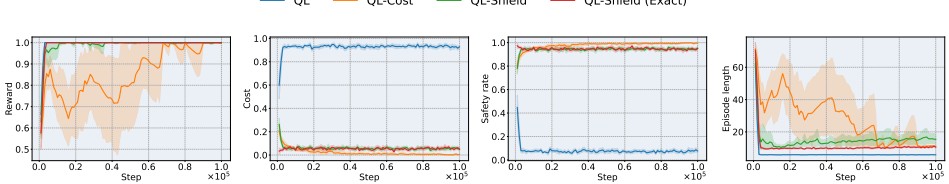

Figure 10: Ablation study with QL-Exact for $9 \times 9$ gridworld property (2).

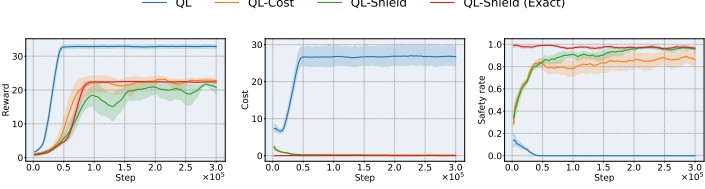

Figure 11: Ablation study with QL-Exact for $15 \times 15$ gridworld property (1).

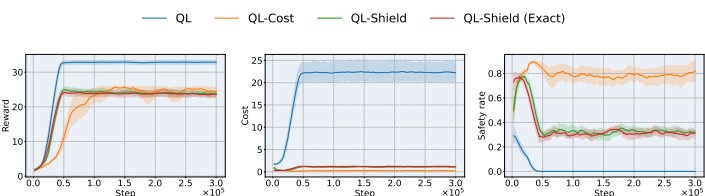

Figure 12: Ablation study with QL-Exact for $15 \times 15$ gridworld property (2).

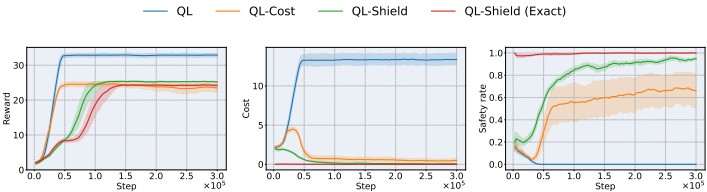

Figure 13: Ablation study with QL-Exact for $15 \times 15$ gridworld property (3).

### D.1 EXTENDED DISCUSSION

In all cases we see that QL-Shield eventually converges to, or close to the safety and task perfor-mance of QL-Exact, which provides a step-wise safety guarantee of $\varepsilon_t$ at the start of training. How-ever, we note that this step-wise safety guarantee doesn't always get us a good episodic guarantee, for example in the Media Streaming environment, QL-Exact immediately provides a step-wise safety guarantee of $1 - \varepsilon_t$, but only provides an 'episodic' safety guarantee of around $0.96$, this is in line with our theory which provides an 'episodic' safety guarantee of $1 - T \cdot \varepsilon_t = 1 - 40 * 0.001 = 0.96$.

### D.2 REWARD SHAPING AND COUNTERFACTUAL EXPERIENCE

In this section we ablate our approach by removing counterfactual experiences (CFE) from our methodology. We note, that this only makes a difference for safety properties that are not simple invariant properties, namely, property (2) and (3) for the $9 \times 9$ *gridworld* and $15 \times 15$ *gridworld* environments. We also experiment with different reward shaping approaches (in addition to CFE), for improving the convergence of the 'backup policy' $\pi_b$. These approaches are detailed below.

**Potential-based reward shaping.** Potential-based reward shaping is an approach used to typically deal with sparse or delayed reward environments. Potential-based reward shaping provides interme-diate rewards of the following form,

$$r'(s', s, a) = r(s', s, a) + \gamma \Phi(s') - \Phi(s) \tag{43}$$

where $\Phi : \mathcal{S} \to \mathbb{R}$ is the potential function. Intuitively this provides the agent with a dense reward signal for moving to 'more promising' states with higher potential values. Note that by discounting the potential values with the original discount factor $\gamma$ this keeps the set of optimal policies that maximize the original reward function unchanged (Ng et al., 1999). Then if the potential function is useful in some sense, this allows us to learn optimal policies more quickly. Icarte et al. (2022) pro-posed automated reward shaping (RS) for reward machines (RM). We adopt this same methodology here. The potential function $\Phi : \mathcal{S} \times \mathcal{Q} \to \mathbb{R}$ is defined over product states $\langle s, q \rangle \in \mathcal{S} \times \mathcal{Q}$, each automaton state $q \in \mathcal{Q}$ is assigned a potential value $v(q)$ computed by running value iteration over the DFA $\mathcal{D}$, with a different discount factor $\gamma' < \gamma$. The cost function for the 'backup policy' $\pi_b$, is then redefined as,

$$\mathcal{C}'(\langle s_t, q_t \rangle) = \mathcal{C}(\langle s_t, q_t \rangle) + \gamma v(q_t) - v(q_{t-1}) \tag{44}$$

**Cycle-based reward shaping.** This approach is inspired by Shah et al. (2024), although Shah et al. (2024) is adapted to LTL-constraints which have a different satisfaction criterion to DFAs, we adapt Shah et al. (2024) to our setting. In particular, we first compute the length shortest path through the DFA $\mathcal{D}$, from each automaton state $q \in \mathcal{Q}$ to the initial state $\mathcal{Q}_0$, we implement this using Breadth-First Search (BFS) in $O(|\mathcal{D}|)$. Each automaton state $q \in \mathcal{Q}$ is assigned a value $v(q)$ which

corresponds to the length of the shortest path from $q$ to $\mathcal{Q}_0$. This incentivizes the agent to return the the initial state $\mathcal{Q}_0$ as quickly as possible, while avoiding the accepting states of the DFA $\mathcal{D}$, which are sink nodes, i.e. $v(q) = \infty \ \forall q \in \mathcal{F}$. The cost function for the 'backup policy' $\pi_b$, is then redefined as,

$$\mathcal{C}'(\langle s_t, q_t \rangle) = \mathcal{C}(\langle s_t, q_t \rangle) + \gamma v(q_t) - v(q_{t-1}) \tag{45}$$

Cycle-based reward shaping is in essence, another instantiation of potential based reward shaping but with different potential values. In practice, setting $v(q) = \infty \ \forall q \in \mathcal{F}$ is infeasible, rather we just ignore the shaped cost function when transitioning to or from an accepting state. We provide the full set of results below; we plot the reward, the cost, the 'episodic' safety rate and the episode length where relevant.

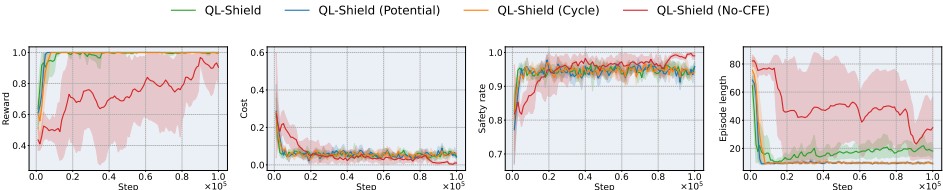

Figure 14: Reward Shaping and CFE for $9 \times 9$ gridworld property (2).

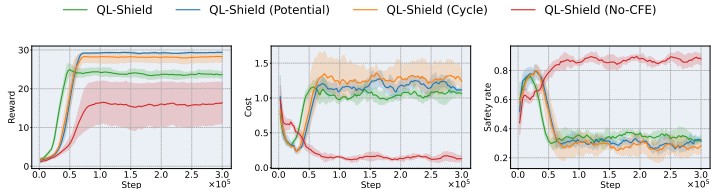

Figure 15: Reward Shaping and CFE for $15 \times 15$ gridworld property (2).

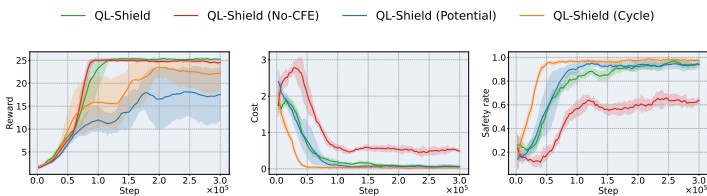

Figure 16: Reward Shaping and CFE for $15 \times 15$ gridworld property (3).

### D.3 EXTENDED DISCUSSION (REWARD SHAPING)

In all cases we clearly see that without CFE QL-Shield has very unstable convergence both in terms of reward and safety, often failing to converge at all to the optimal policy. This demonstrates that CFE is crucial for efficiently learning the safety objective of the environment, when the safety-property is more complex. CFE alleviates the issue of slow and unstable convergence, however for property (2) in both the the $9 \times 9$ and $15 \times 15$ *gridworld*, potential-based reward shaping and cycle-based reward shaping do seem to improve the performance of the agent, with both approaches resulting in more stable convergence to the shortest route through the environment for the $9 \times 9$ gridworld, and an overall higher reward policy for the $15 \times 15$ *gridworld*. However, for property (3) in the $15 \times 15$ *gridworld*, both potential-based reward shaping and cycle-based reward shaping appear to result in slightly unstable learning for the 'task policy' $\pi_r$. We note, that for property (3), cycle-based reward shaping does improve the safety performance compared to the basic QL-Shield with CFE. More investigation and hyperparameter tuning is likely required to understand which approach is the de facto 'best'.

### D.4 ADDITIONAL RESULTS (PACMAN)

In this section we present some additional results, in the Pacman environment from Voloshin et al. (2022). In particular, we provide results for QL, QL-Cost, QL-Shield and QL-Exact (defined earlier). In this environment the task (reward) objective is relatively straightforward (+1 for food), however, balancing this with the safety objective $\Box \neg ghost$ is challenging due to the highly stochastic behaviour of the ghost. We provide the results below; we plot the reward, cost and 'episodic' safety rate.

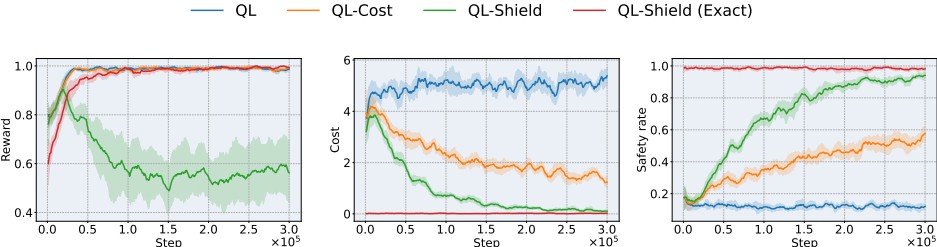

Figure 17: Additional results for Pacman.

### D.5 EXTENDED DISCUSSION (PACMAN)

As expected, when provided with the transition probabilities $\mathcal{P}$ and an optimal safe 'backup policy' $\pi_b^*$, QL-Exact achieves the desired level of safety from the beginning of training. We see also that the shield is not restrictive at all and QL-Exact quickly converges to the optimal reward policy. QL-Shield also quickly converges, but almost never remains safe for the entire episode duration. QL-Cost quickly finds an optimal task policy, and slowly starts to converge to the optimal safe policy, however this convergence is much slower than QL-Shield. We see that QL-Shield converges to near the desired safety level within 300000 timesteps, however the convergence of the 'task policy' appears unstable during training. This environment has $\approx 8000$ states, making Q-learning challenging but still feasible. The slow convergence of the 'safe policy' is likely impeding the progress of the 'task policy'; once the 'safe policy' has properly converged we might expect that the shield overrides the 'task policy' in a more consistent manner from each state, allowing the 'task policy' converge to the expected performance of QL-Exact. Additional hyperparameter tuning of the learning rate for the 'task policy' might also improve matters.

## E HYPERPARAMETERS AND IMPLEMENTATION DETAILS

### E.1 ACCESS TO CODE

To maintain a high standard of anonymity we provide code for both the gridworld and Atari Seaquest experiments in the supplementary material as part of the paper submission. The gridworld environments are implemented with the OpenAI Gym interface (Brockman et al., 2016). Tabular Q-learning is implemented with *numpy* in *Python*, the model checking procedures (both exact and statistical) are implemented with JAX (Bradbury et al., 2018) which supports vectorized computation on GPU and CPU. The code for Atari Seaquest is our own branch of the code base for AMBS (Goodall & Belardinelli, 2023), this also requires JAX among other preliminaries, for setup instructions please refer to the AMBS code base `https://github.com/sacktock/AMBS` (MIT License). For PPO-Lag (Ray et al., 2019) and CPO (Achiam et al., 2017), we use the implementations provided by Omnisafe (Jiaming Ji, 2023), the code for running these benchmarks can also be found in the supplementary material however, for setup instructions please refer to the Omnisafe code base `https://github.com/PKU-Alignment/omnisafe` (Apache-2.0 license).

**Training details.** For collecting both sets of experiments we has access to 2 NVIDIA Tesla A40 (48GB RAM) GPU and a 24-core/48 thread Intel Xeon CPU each with 32GB of additional RAM. For the 'colour' gridworld experiments each run can take several minutes up to a day depending on which property is being tested and whether `exact` or `statistical` model checking is used.

For the Atari Seaquest experiments each run can take 8 hours to 1 day depending on the precise configuration of DreamerV3, in general we see a slow down of $\times 2$ when using DreamerV3-Shield compared to the unmodified DreamerV3 baseline. Memory requirements may differ depending on the DreamerV3 configuration used, for the *xlarge* configuration 32GB of GPU memory will suffice.

**Statistical significance.** Error bars are provided for each of our experiments. In particular, we report 5 random initializations (seeds) for each experiment, the error bars are non-parametric (bootstrap) 95% confidence intervals, provided by `seaborn.lineplot` with default parameters: `errorbar=('ci', 95)`, `n_boot=1000`. The error bars capture the randomness in the initialization of the DreamerV3 world model and policy parameters, the randomness of the environment and any randomness in the batch sampling.

### E.2 THE AUGMENTED LAGRANGIAN

We first define the following objective functions,

$$J_{\mathcal{R}}(\pi) = \mathbb{E}_\pi \left[ \sum_{t=0}^{T} \mathcal{R}(s_t, a_t) \right] \tag{46}$$

$$J_{\mathcal{C}}(\pi) = \mathbb{E}_\pi \left[ \sum_{t=0}^{T} \mathcal{C}(s_t, a_t) \right] \tag{47}$$

$$\tag{48}$$

The augmented Lagrangian (Wright, 2006) is an adaptive penalty-based technique for the following constrained optimization problem,

$$\max_\pi J_{\mathcal{R}}(\pi) \quad \text{subject to} \quad J_{\mathcal{C}}(\pi) \leq d \tag{49}$$

where $d$ is some cost threshold. The corresponding Lagrangian is given by,

$$\max_\pi \min_{\lambda \geq 0} \left[ J_{\mathcal{R}}(\pi) - \lambda \left( J_{\mathcal{C}}(\pi) - d \right) \right] = \max_\pi \begin{cases} J_{\mathcal{R}}(\pi) & \text{if } J_{\mathcal{C}}(\pi) < d \\ -\infty & \text{otherwise} \end{cases} \tag{50}$$

The LHS is an equivalent form for the constrained optimization problem (RHS), since if $\pi$ is feasible, i.e. $J_{\mathcal{C}}(\pi) < d$ then the maximum value for $\lambda$ is $\lambda = 0$. If $\pi$ is not feasible then $\lambda$ can be arbitrarily large to solve this equation. Unfortunately this form of the objective function is non-smooth when moving from feasible to infeasible policies, thus we introduce a proximal relaxation of the augmented Lagrangian (Wright, 2006),

$$\max_\pi \min_{\lambda \geq 0} \left[ J_{\mathcal{R}}(\pi) - \lambda \left( J_{\mathcal{C}}(\pi) - d \right) + \frac{1}{\mu_k}(\lambda - \lambda_k)^2 \right] \tag{51}$$

where $\mu_k$ is a non-decreasing penalty multiplier dependent on the gradient step $k$. The new term that has been introduced here encourages the $\lambda$ to stay close to the previous value $\lambda_k$, resulting in a smooth and differentiable function. The derivative w.r.t $\lambda$ gives us the following gradient update step,

$$\lambda_{k+1} = \begin{cases} \lambda_k + \mu_k(J_{\mathcal{C}}(\pi) - d) & \text{if } \lambda_k + \mu_k(J_{\mathcal{C}}(\pi) - d) \geq 0 \\ 0 & \text{otherwise} \end{cases} \tag{52}$$

At each gradient step, the penalty multiplier $\mu_k$ is updated in a non-decreasing way by using some small fixed (power) parameter $\sigma$,

$$\mu_{k+1} = \max\{(\mu_k)^{1+\sigma}, 1\} \tag{53}$$

The policy $\pi$ is then updated by taking gradient steps of the following unconstrained objective,

$$\tilde{J}(\pi, \lambda_k, \mu_k) = J_{\mathcal{R}}(\pi) - \Psi_{\mathcal{C}}(\pi, \lambda_k, \mu_k)$$

where,

$$\Psi_{\mathcal{C}}(\pi, \lambda_k, \mu_k) = \begin{cases} \lambda_k(J_{\mathcal{C}}(\pi) - d) + \frac{\mu_k}{2}(J_{\mathcal{C}}(\pi) - d)^2 & \text{if } \lambda_k + \mu_k(J_{\mathcal{C}}(\pi) - d) \geq 0 \\ -\frac{(\lambda_k)^2}{2\mu_k} & \text{otherwise} \end{cases}$$

### E.3 TABULAR RL

Table 1: Hyperparameter details for QL, QL-Cost and QL-Shield

| Name | Symbol | Value |
|------|--------|-------|
| **Q-Learning** | | |
| Learning rate | $\alpha$ | 0.1 |
| Discount factor | $\gamma$ | 0.95 |
| Exploration type | - | Boltzmann |
| Temperature | $\tau$ | 0.05 |
| **QL-Shield** | | |
| Model checking type | - | Statistical |
| Number of samples | $m$ | *varies* |
| Step-wise safety | $\varepsilon_t$ | *varies* |
| Failure probability | $\delta_t$ | *varies* |
| Model checking horizon | $N$ | *varies* |
| Approximation error | $\varepsilon'$ | *varies* |
| **'Backup policy'** | | |
| Learning rate | $\alpha$ | 0.1 |
| Discount factor | $\gamma$ | 0.95 |
| Exploration type | - | Boltzmann |
| Temperature | $\tau$ | 0.01 |
| Cost coefficient | $c$ | 10.0 |

For the hyperparameters that vary we provide the following details. For **Media Streaming**: $m = 8000$, $\varepsilon_t = 0.001$, $\delta_t = 0.01$, $N = 5$, $\varepsilon' = 0.02$, **Bridge Crossing**: $m = 8000$, $\varepsilon_t = 0.05$, $\delta_t = 0.01$, $N = 5$, $\varepsilon' = 0.02$, $9 \times 9$ **gridworld**: property (1): $m = 16000$, $\varepsilon_t = 0.01$, $\delta_t = 0.01$, $N = 3$, $\varepsilon't = 0.01$, property (2): $m = 8000$, $\varepsilon_t = 0.12$, $\delta_t = 0.01$, $N = 5$, $\varepsilon' = 0.02$ and for $15 \times 15$ **gridworld**: property (3) $m = 1000$, $\varepsilon_t = 0.001$, $\delta_t = 0.01$, $N = 13$, $\varepsilon' = 0.05$.

For PPO-Lag (Ray et al., 2019) and CPO (Achiam et al., 2017) the only hyperparameters that vary other than the cost threshold $C$ is the steps per epoch $n$. For **Media Streaming**: $n = 400$, **Bridge Crossing** $n = 2000$, $9 \times 9$ **gridworld** $n = 1000$ and for $15 \times 15$ **gridworld** $n = 2500$.

Table 2: Hyperparameter details for PPO-Lag (Ray et al., 2019) and CPO (Achiam et al., 2017) – gridworld environments

| Name | Symbol | Value |
|---|---|---|
| Actor learning rate | $\eta$ | 0.0003 |
| Discount factor | $\gamma$ | 0.95 |
| Cost coefficient | $c$ | 1.0 |
| Cost threshold | $C$ | *varies* |
| Cost gamma | $\gamma_c$ | 0.95 |
| TD-lambda | $\lambda$ | 0.95 |
| Cost TD-lambda | $\lambda_c$ | 0.95 |
| Max grad norm | - | 0.5 |
| Entropy coefficient | - | 0.0 |
| Steps per epoch | $n$ | *varies* |
| **PPO-Lag** | | |
| Initial Lagrangian multiplier | $\lambda_{init}$ | 10.0 |
| Update iterations (per epoch) | $k$ | 40 |
| Epsilon clip | $\epsilon_{clip}$ | 0.2 |
| Batch size | $B$ | 64 |
| **CPO** | | |
| Update iterations (per epoch) | $k$ | 10 |
| Batch size | $B$ | 128 |

## E.4 DEEP RL

Table 3: General hyperparameter details for DreamerV3 (Hafner et al., 2023)

| Name | Symbol | Value |
|------|--------|-------|
| Replay capacity | $D$ | $10^6$ |
| Batch size | $B$ | 16 |
| Batch length | - | 64 |
| Number of envs | - | 8 |
| Train ratio | - | 64 |
| Number of MLP layers | - | 5 |
| Number of MLP units | - | 1024 |
| Activation | - | LayerNorm + SiLU |
| **World Model** | | |
| Configuration size | - | `medium` |
| Number of latents | - | 32 |
| Classes per latent | - | 32 |
| Number of layers | - | 3 |
| Number of hidden units | - | 640 |
| Number of recurrent units | - | 1024 |
| CNN depth | - | 48 |
| RSSM loss scales | $\beta_{\text{pred}}, \beta_{\text{dyn}}, \beta_{\text{rep}}$ | 1.0, 0.5, 0.1 |
| Predictor loss scales | $\beta_o, \beta_r, \beta_c, \beta_\gamma$ | 1.0, 1.0, 1.0, 1.0 |
| Learning rate | - | $10^{-4}$ |
| Adam epsilon | $\epsilon_{\text{adam}}$ | $10^{-8}$ |
| Gradient clipping | - | 1000 |
| **Actor Critic** | | |
| Roll-out horizon | $H$ | 15 |
| Discount factor | $\gamma$ | 0.997 |
| TD lambda | $\lambda$ | 0.95 |
| Critic EMA decay | - | 0.98 |
| Critic EMA regularizer | - | 1 |
| Return norm. scale | $S_{\text{reward}}$ | $\text{Per}(R, 95) - \text{Per}(R, 5)$ |
| Return norm. limit | $L_{\text{reward}}$ | 1 |
| Return norm. decay | - | 0.99 |
| Actor entropy scale | $\eta_{\text{actor}}$ | $3 \cdot 10^{-4}$ |
| Learning rate | - | $3 \cdot 10^{-5}$ |
| Adam epsilon | $\epsilon_{\text{adam}}$ | $10^{-5}$ |
| Gradient clipping | - | 100 |

Table 4: Hyperparameter details for DreamerV3-Lag

| Name | Symbol | Value |
|------|--------|-------|
| Penalty multiplier | $\mu_k$ | $5 \cdot 10^{-9}$ |
| Initial Lagrange multiplier | $\lambda^k$ | 0.01 |
| Penalty power | $\sigma$ | $10^{-6}$ |
| Cost coefficient | $C$ | 1.0 |
| Cost threshold | $d$ | 1.0 |

Table 5: Hyperparameter details for DreamerV3-Shield

| Name | Symbol | Value |
|------|--------|-------|
| Number of samples | $m$ | 512 |
| Step-wise safety | $\varepsilon_t$ | 0.01 |
| Failure probability | $\delta$ | 0.01 |
| Lookahead/shielding horizon | $N$ | $\{30, 50\}$ |
| Approximation error | $\varepsilon'$ | 0.01 |
| Cost coefficient ('backup policy') | $c$ | 10 |

Table 6: Hyperparameter details for PPO-Lag and CPO – Atari Seaquest environment

| Name | Symbol | Value |
|------|--------|-------|
| Actor learning rate | $\eta$ | 0.00003 |
| Discount factor | $\gamma$ | 0.9967 |
| Initial Lagrangian multiplier | $\lambda_{init}$ | 10.0 |
| Cost coefficient | $c$ | 1.0 |
| Cost threshold | $C$ | 1.0 |
| Cost gamma | $\gamma_c$ | 0.95 |
| TD-lambda | $\lambda$ | 0.95 |
| Cost TD-lambda | $\lambda_c$ | 0.95 |
| Max grad norm | - | 40.0 |
| Entropy coefficient | - | 0.0 |
| Steps per epoch | $n$ | 20000 |
| **PPO-Lag** | | |
| Update iterations (per epoch) | $k$ | 40 |
| Epsilon clip | $\epsilon_{clip}$ | 0.2 |
| Batch size | $B$ | 64 |
| **CPO** | | |
| Update iterations (per epoch) | $k$ | 10 |
| Batch size | $B$ | 128 |

