# OpenReview forum: "Lookahead Shielding for Regular Safety Properties in Reinforcement Learning"
_ICLR.cc/2025/Conference — Submitted to ICLR 2025_

### Official Review · Reviewer_kJrR · 2024-10-30

**Soundness:** 3
**Presentation:** 3
**Contribution:** 3
**Rating:** 6
**Confidence:** 4

**Summary:**

The paper tackles the problem of safety in Reinforcement Learning by shielding. The core concept of shielding is to have an agent learn a policy while preventing it from taking harmful actions. In this paper, a new shielding method is proposed where two policies are learned separately: One to maximize the reward, and another to maximize the probability the agent stays in a set of "probabilistic safe states". The combined approach then follows the former policy while switching to the latter if the agent would leave the aforementioned probabilistic safe states.

**Strengths:**

While the core idea of shielding has been explored quite a lot recently, and many of the concepts used in the paper (such as proabilistic safe sets) are known in the literature, combining the methods in this way has (to the best of my knowledge) not been considered yet. The approach comes with a probabilistic (step-wise) safety guarantees that are computable and importantly only come with very mild assumptions (such as knowing the number of states) compared to previous safe appraches requiring additional information such as the entire transition structure. An unavoidable restriction is necessary is to assume the safe backup policy to be learned correctly (in particular that it identified the probabilistic safe states correctly), however in practice there is no way around this with this framework. This limitation is adressed in the appendix where the shielding approach is once provided with the correct backup policy and once has to be learned, with the results showing convergence to a near-optimal backup policy within relatively few samples.

The paper starts with an extensive section on the formal background which provides all necessary background knowledge to make the paper self-contained. The paper is mostly strucured and it is easy to follow the main concepts in the paper. Proofs for all statements are provided in the appendix. Related work is thoroughly discussed at the end of the paper.

I was especially impressed with the experimental evaluation in which the new proposed method outperforms state-of-the-art approaches in accumulated reward while simultaneously showing equally safe or even safer behaviour throughout the different examples. This holds for boith the tabular and deep RL setting. The implementation is also fully provided in the supplementary meterial. The appendix contains further details and insights of the experiment, including all hyperparameters used.

**Weaknesses:**

The main weaknesses of the paper are imprecisions in some formal sections.

Although the theoretical background is rigorously defined for the most part, it unfortunately lacks precision at some of the most important places: In Eq. 1 the shielding strategy -- the core contribution of the paper -- is defined. However, the definition is ambiguous: the term $Pr(⟨s, q⟩ \models \lozenge^{\leq N} accept)$ is not defined since $Pr$ was only defined on Markov Chains, not MDP. It is not entirely clear which policy is used to resolve the non-determinism in the MDP: $\pi_{safe}$ or $\pi_{safe}^N$ or the minimizing policy (from my understanding these are not necessarily the same if $\pi_{safe}$ is learned incorrectly)? Also the "(given $a$)" part is ambiguous -- I suppose from context that this means you actually resolve the non-determinism by performing one step according to $\pi_{task}$, and afterwards accoring to some safe policy. This should be clarified.

A similar question arises with the definition of the backup policy $\pi_{safe}$: In Remark 3.2 it is exaplained that the objective $\pi_{safe}$ is ought to minimize is non-markov, however only memoryless policies are defined and the paper. Also the time step is not an explicit input in the definition of $\pi_{shield}$ -- does this mean you restrict $\pi_{safe}$ to memoryless strategies?

For assumption 3.8 it is not clear how this is supposed to be parsed. It starts by "For all irrecoverable actions $a\in A$ ..." but then never uses $a$. Also I would expect this to be an implication, i.e. "if $Pr(...) > \varepsilon$ then $N\geq N^{*}$" from context, but I am not sure here.

---

While the experimental section is convincing and I have no reason to doubt the results, for reproduction of the results a proper explanation of how to setup the the experiments (e.g. required packages) and how to obtain the results in the paper and use the provided code (e.g. as a README) is instrumental. Ideally, a docker container or similar should be provided to avoid issues with other platforms and user setups.

For the presentation, I suggest emphasizing the separation from existing methods, in particular extending section 3.2. On the technical side this is quite clear since the CMDP approach is also formally defined here, however an intuition is missing here. For readers it would be insightful to know what the core difference behind the approaches are and why/for which kinds of models one would expect this shielding approach to outperform others such as CMDP.

While Theorem 3.10 provides theoretical guarantees on the safety it is not clear how practical these are, i.e. if the required number of samples is reasonable, or astronomically large. An application of Thm. 3.10 to the experimental section would be also be desirable discuss the tightness of the guarantees, i.e. whether there is a substantial gap between guaranteed safety and observed safety.

As a minor comment, Theorem 3.10 is highly related to [R3], where the problem of relating the number of samples to the probability that Eq. (5) holds is tackled directly. This should be cited. However, I also want to mention that their results are only an improvement if a bound on the branching factor is known (other than the trivial bound of number of states), see [R4].

**Questions:**

I have some questions for understanding Definition 3.1:

You say you train the backup policy with the goal of minimizing discounted expected cost. (l. 206) However this introduces a non-linear weighting of costs which may lead to the backup policy preferring some policy $\pi_1$ over $\pi_2$, even though the expected number of steps until reaching appect might be higher in $\pi_2$. **Why do you chose the discounted cost over the undiscounted cost which is easily learnable in the finite-horizon setting?** (Remark: In the infinite horizon case it would also not be necessary to introduce the $\omega$-automaton in Definition 3.1 since optimizing for your objective is equivalent to minimizing the (discounted) expected number of steps until the accept state in a single run, correct?)

In Remark 3.2 you write "It is important to note that for regular safety properties the corresponding cost function is defined over the product states and is thus non-Markov", implying the product Markov chain is responsible for the cost function being non-Markov. However, **isn't the finite horizon the culprint of making the cost function non-Markov**, i.e. would the cost function be Markov if you defined it as the infinite-horizon objective?

In my mind, using the infinite-horizon cost as objective (either discounted, or long-run average cost [aka mean payoff]) would be a more reasonable choice. In the finite horizon setting you specifically allow the backup policy to lead to unsafe behaviour, as long as it can guarantee it happens after the time horizon. This is highly undesirable in many practical settings. **How do you justify choosing the finite-horizon cost as the objective?**

---

Other questions:

* Can you clarify the ambiguities I pointed out in the "Weaknesses" section, i.e. (1) Which policy is used to resolve the non-determinism when determining whether to use $\pi_{task}$? (2) Can you formally states how "(given a)" is to be interpreted? Does the resolving policy always pick $a$ in $s$ or only for one step? (3) Is $\pi_{safe}$ memoryless? (4) Can you clarify Assumption 3.8?
* The shielding policy is presented as a binary choice between the preferred maximizing action and the safe backup action. Since usually an RL agent evaluates all action, it can usually also rank all actions. Have you considered extending this to a setting where the action with the highest ranking is chosen, restricted to those actions which satisfy the condition in Eq. (1)?
* Can you provide an example of the theoretical bounds of Thm. 3.10 applied to some examples in your experimental setup? How far are the bounds from the actual rate of unsafe behaviour observed?

---

minor comments

* the paper uses a mix of (mostly) AE and (some) BE, e.g. "minimize" vs. "maximise" and i.e. sometimes followed by a comma and sometimes not
* for intuitive understanding of safety properties, it might be beneficial to give the accepting state a negative name (e.g. "fail") instead of accept
* line 168: $\pi_{safe}$ is not necessarily low-reward
* line 237: "...is modified slightly to update give zero reward..." - I guess either "give" or "update" is too much here
* for the statistical model checking approach, I would suggest more recent citations that provide methods that are more sample efficient than Hoeffding's inequality, e.g. [R1,R2]
* for understanding, it would be helpful to define probabilistic safe sets (Def 3.6) earlier since you already mention them a lot in section 3
* l. 326: " Intuitively the ‘backup policy’ πsafe defines an (T -step) probabilistic safe set" -- the definion of the probabilistic safe set is independent of $\pi_{safe}$. Actually it is the other way around that $\pi_{safe}$ depends on the probabilistic safe set.

[R1] Jegourel, Cyrille & Sun, Jun & Dong, Jin. (2019). Sequential Schemes for Frequentist Estimation of Properties in Statistical Model Checking. ACM Transactions on Modeling and Computer Simulation. 29. 1-22. 10.1145/3310226.

[R2] Meggendorfer, T., Weininger, M., & Wienhöft, P. (2024). What Are the Odds? Improving the foundations of Statistical Model Checking. ArXiv, abs/2404.05424.

[R3] Weissman, T., Ordentlich, E., Seroussi, G., Verdú, S., & Weinberger, M.J. (2003). Inequalities for the L1 Deviation of the Empirical Distribution.

[R4] Wienhoft, P., Suilen, M., Simão, T.D., Dubslaff, C., Baier, C., & Jansen, N. (2023). More for Less: Safe Policy Improvement With Stronger Performance Guarantees. International Joint Conference on Artificial Intelligence.

---

> ### Author Response · Authors · 2024-11-18
> **Response #1**
>
> We thank the reviewer for taking the time to read our paper and for providing such an in-depth assessment of our approach, we appreciate that they were able to see the merit of our contributions
>
> ## Re Weaknesses:
>
> Re README: apologies for this, we thought our supplementary material included READMEs, this was an oversight when submitting, the supplementary materials should now be updated to include README files.
>
> Thanks for bringing [R3] to our attention, indeed we will aim to include this in our citation list as it seems relevant, interestingly I believe we mentioned the conservativeness of our bound given the dependence on $|S|$ which could be replaced by the maximum number of successor states if this is known, or equivalently an upper bound on the branching factor as the reviewer rightly points out. If time allows we will aim to improve the bound provided in Theorem 3.10 given the references you have provided.
>
> ## Re Questions:
>
> "I have some questions for understanding Definition 3.1:"
>
> This is a very small technical detail we missed, thanks for bringing it to our attention. Please correct me if I am wrong, but I believe there is a slight conflation between the cost function (Definition 3.1) and the objective function $ \mathbb{E}\_{\pi} [ \sum^{T}\_{t=0} \gamma^t \mathcal{C}(s\_t, q\_t) ] $ which the safe (backup) policy seeks to minimize, we claim that the cost function $\mathcal{C} : \mathcal{S} \times \mathcal{Q} \to \mathbb{R}$ is non-Markov, since the automaton state $q\_t$ at timestep $t$ could depend on some arbitrary history of past states, the cost function is Markov on the product state space $\mathcal{S} \times \mathcal{Q}$ however. On the other hand, the objective function $\mathbb{E}\_{\pi}[\sum^{T}\_{t=0}\gamma^t \mathcal{C}(s\_t, q\_t)]$ is written incorrectly. We update the $\pi\_{\textit{safe}}$ with the following Q-learning update rule,
>
> $$\hat{Q}\_{\textit{safe}}(s\_t, q\_t, a\_t) \stackrel{\alpha}{\longleftarrow} \gamma \max\_{a} \{ \hat{Q}\_{\textit{safe}}(s\_{t+1}, q\_{t+1}, a) \} - \mathcal{C}((s\_t, q\_t))$$
>
> This will give us a policy that minimizes the **infinite horizon discounted cost**, thus the objective function should in fact be written as $\mathbb{E}\_{\pi}[\sum^{\infty}\_{t=0}\gamma^t \mathcal{C}(s\_t, q\_t)]$, we agree that the discounted finite objective function poses issues. We hope this clears up any confusion, please feel free to ask for further clarification.

---

> ### Author Response · Authors · 2024-11-18
> **Response #2**
>
> Q: *"Can you clarify the ambiguities I pointed out in the "Weaknesses" section, i.e. (1) Which policy is used to resolve the non-determinism when determining whether to use? (2) Can you formally states how "(given a)" is to be interpreted? Does the resolving policy always pick in or only for one step? (3) Is memoryless? (4) Can you clarify Assumption 3.8?"*
>
> A: We agree with the particular point highlighted by the reviewer here, it is not formally stated which policy/actions are used to resolve the non-determinism of the MDP. Although, I think the reviewer has correctly identified that (given $a$) means we first resolve the non-determinism of the MDP with the action $a$ sampled from $\pi_{\textit{task}}$ and then with $\pi_{\textit{safe}}$ thereafter, we will try to think of a way to formally state this in the revised version of our paper. Importantly, we note that this does not mean $a$ is always picked from $s$ just for the first step in resolving the non-determinism.
>
> Technically speaking we did only formally introduce memory-less policies over the state space $\mathcal{S}$, indeed the objective that $\pi_{\textit{safe}}$ seeks to minimize is non-Markov with respect to the the state space $\mathcal{S}$, however $\pi_{\textit{safe}}$ operates on the product state space $\mathcal{S} \times \mathcal{Q}$ thus $\pi_{\textit{safe}}$ is memory-less on $\mathcal{S} \times \mathcal{Q}$ and should be defined as a function $\pi_{\textit{safe}} : \mathcal{S} \times \mathcal{Q} \to \textit{Dist}(\mathcal{A})$, we will aim to make this more explicit. Regarding the model checking horizon $N$, this is fixed globally and does not need to be an input to $\pi_{\textit{shield}}$.
>
> In Assumption 3.8 the probability $\Pr(\ldots)$ is interpreted as before, by first using $a \in \mathcal{A}$ to resolve the non-determinism of the MDP and $\pi_{\textit{safe}}$ thereafter. I think the confusion here can be resolved by implementing some of the changes mentioned above. In general, Assumption 3.8 is supposed to guarantee that there exists some optimal model checking horizon $N^*$ which we can use to identify all irrecoverable actions, $N \geq N^*$ means the actual model checking horizon $N$ that we have chosen is larger than this optimal model checking horizon, so no implication is needed here, we can try re-word this assumption to make it more clear what we mean.
>
> Q: *"The shielding policy is presented as a binary choice between the preferred maximizing action and the safe backup action. Since usually an RL agent evaluates all action, it can usually also rank all actions. Have you considered extending this to a setting where the action with the highest ranking is chosen, restricted to those actions which satisfy the condition in Eq. (1)?"*
>
> A: This is an excellent point, we have thought about this, although we haven't actually experimented with this, in theory we could rank the available actions by the task policies Q values and then iteratively check whether each action is safe or not starting from the highest ranked action and once we find a safe action we use it. For small action spaces, like grid worlds, this seems like a reasonable idea, for large action spaces, this might incur significant overhead, for example Seaquest has $|\mathcal{A}| = 18$ which might be impractical.
>
> Q: *"Can you provide an example of the theoretical bounds of Thm. 3.10 applied to some examples in your experimental setup? How far are the bounds from the actual rate of unsafe behaviour observed?"*
>
> A: For simplicity we will choose the Media streaming environment to demonstrate how our bounds hold up. In this experiment we have, $|\mathcal{S}| = 20$, $|\mathcal{A}|=2$, and we set, $\varepsilon_t = 0.001$, $N=5$, $\varepsilon'=0.02$, $\delta_t=0.01$, even using the bound provided in the appendix of [R4], which considers a known branching factor $k$ (for media streaming $k=3$), the bound appears to be unwieldy, requiring that each state-action pair be visited $O(10^7)$ times.
> $$m \approx \frac{32 \cdot N^2}{\varepsilon'} \log (\frac{2|\mathcal{S}|^2 |\mathcal{A}^2|2^k}{(k-1)\delta_t/2})$$
> Nevertheless, we obtain a a safety rate of $1 - \sum^{40}\_{t=1} \varepsilon\_t = 0.96$ with probability $1 - \sum^{40}\_{t=1}\delta\_t = 0.6$. Studying Figure 6 (Appendix C) closely, we see that this safety rate is empirically achieved by QL-Shield (Exact) and we see that our approach (QL-Shield) quickly converges to the safety rate achieved by QL-Exact. Please let me know if you think I've done my calculations for this wrong or perhaps bounds such as this are always so unwieldy?
>
> We thank the reviewer for their additional minor comments, they seem very helpful and we will aim to incorporate them in to a revised version of our paper.

---

> ### Author Response · Authors · 2024-11-25
>
> Once again we thank the reviewer for their comprehensive review and for finding the time to respond to our rebuttal. We understand the reviewer still has some reservations, particularly considering the sample complexity bound. Our aim is to upload a revised version of the paper before the end of the discussion period, that implements some of the changes mentioned in our initial response. We then invite the reviewer to consider adjusting their score as they see appropriate. Thanks again.

---

### Official Review · Reviewer_4KfC · 2024-10-30

**Soundness:** 3
**Presentation:** 4
**Contribution:** 3
**Rating:** 6
**Confidence:** 4

**Summary:**

This paper presents "lookahead shielding," a framework for reinforcement learning with regular safety properties that requires minimal prior knowledge compared to previous shielding approaches. The framework operates by dynamically switching between a reward-maximizing "task policy" and a safety-focused "backup policy" based on probabilistic safety assessments using finite horizon model checking with learned environment dynamics. The paper provides theoretical analysis comparing the approach to constrained Markov decision process (CMDP) methods and proves safety guarantees for the tabular setting under reasonable assumptions. Through experiments in both tabular environments (media streaming, bridge crossing, and gridworld variants) and the Atari environment using deep RL, the paper demonstrates empirical effectiveness compared to baseline approaches like PPO-Lagrangian and constrained policy optimization (CPO). The key innovation lies in handling regular safety properties (which are non-Markovian) while requiring minimal prior knowledge of the environment, unlike previous approaches that demanded strong assumptions about environment dynamics or topology.

**Strengths:**

The paper reformulates the safety-constrained RL problem by separating concerns between a task policy and a backup policy, while introducing a dynamic switching mechanism based on probabilistic safety assessments. Also, the paper presents an integration of finite-horizon model checking with learned dynamics. It handles both exact and statistical model checking approaches, providing error bounds and computational complexity analysis for each case. The paper provides a treatment of regular safety properties, which are inherently non-Markovian. The approach of using deterministic finite automata (DFA) for specification and product construction for verification, while maintaining computational tractability, is interesting. Furthermore, the framework demonstrates sample efficiency through its use of counterfactual experiences in the backup policy training and integration of model-based planning with learned dynamics.

**Weaknesses:**

- The theoretical guarantees rely heavily on Assumptions 3.8 and 3.9, which are quite strong. Assumption 3.8 requires that any irrecoverable actions can be identified within a finite horizon N*, but there's no practical method provided to determine this horizon.
- The product construction between the MDP and safety automata can lead to state space explosion. While the paper briefly mentions this issue, it doesn't provide a solution. For large DFAs or complex safety properties, the product state space could become intractable.
- The safety guarantees rely on union bounds across timesteps, which could be overly conservative in practice. For long episodes, this could lead to conservative behavior or require small per-time step violation probabilities.
- The need for repeated model-checking during execution could introduce significant computational overhead, especially in real-time applications.
- For complex safety properties, constructing appropriate DFAs could be challenging, and the paper does not address automated methods for DFA construction or optimization. The size of the DFA directly impacts computational complexity.

**Questions:**

- The union bound for episodic safety seems very conservative. Have you explored tighter bounds or alternative approaches? For instance, could martingale-based techniques provide tighter guarantees?
- How can we verify Assumption 3.8 (existence of finite N* for identifying irrecoverable actions) in practice? Could you provide a method or heuristic for estimating N*?
- For large DFAs representing complex safety properties, the product state space becomes intractable. Have you explored hierarchical or compositional approaches to manage this complexity?
- The backup policy seems crucial for performance. Could you elaborate on alternative designs for the backup policy beyond the current cost-based approach?
- Could you provide more details about the real-time performance requirements? What is the typical computation time for model-checking in your deep RL experiments?

---

> ### Author Response · Authors · 2024-11-18
> **Re Questions**
>
> We thank the reviewer for taking the time to read our paper paper and ask some very thoughtful questions.
>
> For our response we will first aim to clear up the reviewers questions and we think this should resolve some the perceived limitations of our approach.
>
> ## Re Questions:
>
> - The union bound for episodic safety seems very conservative. Have you explored tighter bounds or alternative approaches? For instance, could martingale-based techniques provide tighter guarantees?
>
> We agree that the union bound does seem overly conservative in practice, as in general we may often not be in a safety-critical state. Although, we note that these results are in line with prior work [1] that also use union bounds for probabilistic safety guarantees. Regrettably we have not explored alternative approaches for tighter bounds, although martingale-based techniques do sounds like a very useful suggestion. In particular, we could use Azuma-Hoeffding to try and bound the expected discounted cost objective of the full shielded policy, this could then offer tighter bounds on the probability of satisfaction of the safety properties, if time permits we will try and see if such an analysis is possible and include it in the revised version of our paper.
>
> [1] "Safe Reinforcement Learning via Statistical Model Predictive Shielding." by Bastani, Osbert, Shuo Li, and Anton Xu.
>
> - How can we verify Assumption 3.8 (existence of finite N* for identifying irrecoverable actions) in practice? Could you provide a method or heuristic for estimating N*?
>
> The choice of $N^*$ is crucial for our safety guarantees, knowing something about the environment can help to choose $N^*$ appropriately, in general we would want $N^*$ to be the longest path from any given state to the probabilistic safe set of the backup policy, in a self-driving car example $N^*$ could be the longest time it takes the car to decelerate to a stationary position, as a heuristic, slightly more than the shortest path in the DFA from the initial state to the accepting state is a good choice that we used in our experiments. For more details we point the reviewer to [2] where, to the best of our knowledge, this assumption appeared for the first time (there is a probabilistic interpretation in the appendix of [2]).
>
> [2] "Safe reinforcement learning by imagining the near future" by Thomas, Garrett and Luo, Yuping and Ma, Tengyu
>
> - For large DFAs representing complex safety properties, the product state space becomes intractable. Have you explored hierarchical or compositional approaches to manage this complexity?
>
> In general, we deemed the synthesis of DFA from complex safety properties beyond the scope for this paper. We assume a DFA is already provided which corresponds to some regular safety property. We note that, for a fixed MDP, the size of the product state space only grows linearly in the size of the DFA. Although, indeed for large DFA and MDP this can get out of hand. Our model checking approach which is based on statistical model checking, mostly alleviates this issue by computing the product states on-the-fly, meaning the (possibly) costly full product does not need to be computed. The only restriction really is if we can fit the DFA in memory, which is usually the case for properties of interest.
>
> - The backup policy seems crucial for performance. Could you elaborate on alternative designs for the backup policy beyond the current cost-based approach?
>
> One could generally use any well-tuned RL algorithm for learning the backup policy online using the cost function. Other works [2] consider LQR to construct a backup policy before hand, although this is not relevant for discrete state spaces, in our ablations in Appendix C, we use value iteration to pre-compute the backup policy. Sound value iteration is used by STORM model checker, interval-bound value iteration can also be used to compute safe policies, although these approaches are only possible when we have access to the precise dynamics of the MDP ahead of time, we make no such assumptions in our paper.
>
> - Could you provide more details about the real-time performance requirements? What is the typical computation time for model-checking in your deep RL experiments?
>
> Model checking is relatively fast, we use JAX and JIT compilation to implement statistical model checking which can generate thousands of samples per second. On good hardware we were able to achieve at least 5 FPS (interactions with the environment) for the most complex safety property (15x15 grid world property (3)) and much quicker for the simpler properties. In the deep RL experiments, model checking during learning resulted in a slow down of x2 -- from 400 down to 200 FPS. We acknowledge that 5 FPS for real-time performance may not necessary be brilliant, however once the dynamics model and safe (backup) policy are fixed we can do much better (discussed in the next comment).

---

> > ### Author Response · Authors · 2024-11-28
> > **Revised Paper**
> >
> > We have now uploaded a revised version of the paper. Beyond some minor changes to notation, we have provided a short discussion on the limitations of our approach, in particular for choosing the model checking horizon $N^*$. We have also added additional experiments to the appendix. We kindly invite the reviewer to read the new material that they find interesting, and we invite them to ask any further questions they may have or adjust their score as they see appropriate.

---

> > > ### Comment · Reviewer_4KfC · 2024-12-02
> > > **All Concerns Addressed**
> > >
> > > Thank you for your response. All my concerns have been resolved. I elevated my score.

---

> ### Author Response · Authors · 2024-11-18
> **Re Weaknesses**
>
> *"The product construction between the MDP and safety automata can lead to state space explosion. While the paper briefly mentions this issue, it doesn't provide a solution"*
>
> This is typically not the case. In our paper we use statistical model-checking which alleviates this issue of "state space explosion" by computing the product states on-the-fly rather than computing the full product. Also, for a fixed size MDP the size of the product is linear in the size of the safety automata.
>
> *"The need for repeated model-checking during execution could introduce significant computational overhead, especially in real-time applications"*
>
> The real time performance of 5 FPS during training may not be practical, however, we only need to repeatedly model check since the dynamics model and safe (backup) policy are non-stationary, i.e. being updated with new experience. Once training has completed and the dynamics model and safe (backup) policy are fixed, we can simply run one entire pass over each state-action pair and check if it is safe, we can then save these results in a hash-map or lookup-table and quickly look them up during deployment with minimal overhead. This is what we do for our ablation in Appendix C, and we enjoyed operation speeds of 3000 FPS.

---

> ### Author Response · Authors · 2024-11-25
> **Remaining Issues**
>
> If the reviewer has no more questions or comments at this time, we will deem, for the most part, the issues raised by the reviewer as resolved, and we will touch upon some of our response in the revised version of our paper, that will be uploaded before the end of the discussion period. In any case we encourage the reviewer to adjust their score as they see appropriate.

---

### Official Review · Reviewer_dvFE · 2024-11-04

**Soundness:** 2
**Presentation:** 2
**Contribution:** 1
**Rating:** 3
**Confidence:** 4

**Summary:**

The paper presents an approach to safe RL in which a 'shielding' style-approach is used, where absolute violations of constraints can be considered in contrast to CMDP-style approaches which consider violations of constraints in expectation over trajectories. The approach leverages a learned dynamics model (using prior work on world model learning e.g. DreamerV3) and learns two policies on-the-fly during training: one that is performant with respect to the reward function of the MDP, and one that will minimize (absolute) constraint violations in the form of a PCTL property. The "safe" policy is trained by seeking to minimize constraint violations that is cast in the form of a DFA that penalizes violating prefixes for the safety property. The approach outperforms baselines that consider CMDP approaches in both tabular and deep RL experiments.

**Strengths:**

The area of safe RL, and the consideration of a broad class of absolute constraints is an important and challenging problem. Bringing more modern learning-based techniques into the fold of safe RL approaches is valuable and I am glad the authors are trying to modernize more classical techniques like shielding by acknowledging the current limitations and applicability of the space. The experimental results show that CMDP approaches, as expected, are unable to satisfy the safety properties while achieving reward-performant policies.

**Weaknesses:**

First and foremost, the writing of the paper has a large number of typographical errors and clarity issues regarding the phrasing and wording of particular sentences and would heavily benefit from a thorough editing pass.

From the introduction of the paper, I was not fully clear on exactly how this work would build on prior efforts and be distinct from the (very large) body of work on addressing logical or absolute constraints. Ultimately, my understanding is that existing works do not usually try to learn the safe "fallback" policy, and some works make prior assumptions on dynamics (although this isn't really true, most works nowadays do not make such assumptions.)

The theoretical guarantees made in the work are fairly conservative, which the authors acknowledge and I appreciate that. However, the guarantees made here also rely on the quality of the *learned* dynamics model and safe policy - in practice, we'll rarely know for sure that these learned components are 'good enough'. As a result, I would want to see a number of ablations in the experimental results to understand how and when the approach will work as expected, and if not, what those circumstances are. For example, how complicated do the safety properties need to be for the approach to start breaking down? As mentioned in remark 3.2, large DFAs that make convergence slow may hurt the approach's efficacy, even with counterfactual ER. When does this happen? Also, DFA or automaton-guided learning for temporal logic properties are *very* well-explored. Can we explore different choices of reward shaping or learning for the safe policy based on this?

A smaller point: using CER relies on Q-learning approaches, which are pretty generally worse than PPO/SAC for deep RL. This will limit empirical success. Prior works have developed logical CER approaches for PPO (see Eventual Discounting Temporal Logic Counterfactual Experience Replay from Voloshin et. al. 2023) that can be used.

Overall, from a contribution standpoint, I don't find that the step forward from existing works is significant, and the novel components of the work are not explored enough to highlight them. I think the interesting parts of using a DFA as a means to learn cost-minimizing policy and the relationship between the switching policy, the learned dynamics model, and the safe policy are not fully ablated in experimental results which makes it hard to fully appreciate the proposed approach. A more thorough distinction and description of related work would help the work a lot - the related works section in the main text does not highlight the technical differences enough.

Additional comments: Section 3.2 that relates the approach to CMDPs isn't totally necessary, it's pretty clear the two things are different. This section, in my opinion, can be struck and the extra space can be used to expand on the points I mentioned in previous paragraphs. There are some potential errors in the preliminary definitions (i.e. line 84: a state, action transition should map to just a single distribution over states instead of a distribution over distributions) that can be cleaned up.

**Questions:**

My questions are also mentioned in the previous section.

* How do different choices of reward shaping or learning for the safe policy based affect performance?
* How complicated do the safety properties need to be for the approach to start breaking down?
* Have you tried comparing your approach to previous approaches to temporal-logic constrained policy optimization e.g. [1] and [2], rather than just CMDP-based approaches?

[1] Policy Optimization with Linear Temporal Logic Constraints. Voloshin et. al. 2022.
[2] LTL-Constrained Policy Optimization with Cycle Experience Replay. Shah et. al. 2024.

---

> ### Author Response · Authors · 2024-11-18
> **Re Weaknesses**
>
> We thank the reviewer for providing a thorough assessment. We will take the time necessary to iron out any typographical errors and issues with phrasing etc.
>
> ## Re Weaknesses:
>
> We appreciate that the reviewer finds some merit in our contributions and recognizes that separately learning a safe (backup) policy is a unique property of our approach. In response to *"However, the guarantees made here also rely on the quality of the learned dynamics model and safe policy - in practice, we'll rarely know for sure that these learned components are 'good enough'"*, we point the reviewer to the theoretical contributions in our paper (Theorem 3.10) which give a precise bound on when we know that our learned dynamics model is good enough for accurate model checking. In particular, after each state-action pair has been visited at least $\mathcal{O}\left(\frac{N^2|\mathcal{S}|^2}{{\varepsilon'}^2} \log\left( \frac{|\mathcal{A}||\mathcal{S}|^2}{\delta'}\right)\right)$ times, then dynamics model is 'good enough' for our purposes, we can easily monitor the number of visits during training. Verifying if the backup policy is 'good enough' is trickier, however given an $\epsilon$-accurate MDP one could run value iteration to compute an $\epsilon$-optimal backup policy, with respect to the expected discounted cost objective. We encourage the reviewer to read Appendix C, where we ablate our approach, by providing the model checker with a perfect model of the MDP and a perfect backup policy (computed with value iteration before training). In this instance we can provide safety guarantees in line with Theorem 3.10 from the beginning of training, empirically we see that our approach here (QL-Shield) quickly matches the performance of (QL-Exact) demonstrating that the dynamics model and backup policy have successfully been learnt.
>
> ### Re smaller point:
>
> For the tabular RL setting, Q-learning has consistently proved to be the most fruitful approach. We agree with the reviewer that in the deep RL setting, Q-learning based approaches have limitations, for our deep RL experiments we do not use Q-learning and leverage DreamerV3 for both policy optimization and dynamics learning. For CMDPs and safe RL more generally model-based approaches have been shown to dramatically outperform model-free approaches based on PPO/TRPO, which is why we opted for DreamerV3, see [1,2].
>
> [1] "Constrained policy optimization via bayesian world models" by As, Yarden and Usmanova, Ilnura and Curi, Sebastian and Krause, Andreas
>
> [2] "Safe DreamerV3: Safe Reinforcement Learning with World Models" by Huang, Weidong and Ji, Jiaming and Zhang, Borong and Xia, Chunhe and Yang, Yaodong
>
> ### Re Additional Comments:
>
> We felt it necessary to include the technical comparison to CMDP-based approaches given that we compare these approaches in our experiments. From a technical stand point it is clear that CMDP-based approaches struggle to satisfy absolute or high probability constraints, as is further evidenced by our experimental results. However, we felt that this was a fair comparison given that, like our approach, CMDP-based approaches also assume no prior knowledge of the transition dynamics of the MDP, no safe backup policy or any other restrictive assumptions.
>
> With regards to line 84 we do not quite see how this is a technical mistake, we defined $\textit{Dist}(\mathcal{S})$ as the set of distributions over $S$, writing $\mathcal{P}: \mathcal{S} \times \mathcal{A} \to \textit{Dist}(\mathcal{S})$ is standard notation that means that the function $\mathcal{P}$ maps from the set of state action pairs to distributions over states, which is the desired interpretation.

---

> > ### Comment · Reviewer_dvFE · 2024-11-26
> > **Re Weaknesses Response**
> >
> > Thanks to the authors for their response. I do better understand the spirit of the contribution, but I still feel that the writing does not make fully clear the advancements from the state-of-the-art. In addition, I don't think the experiments highlight the novelty of the approach enough and I don't see the baseline comparisons as compelling enough to justify changing my score.
> >
> > Re: line 84, line 75 defines Dist as a set of distributions, which is likely where the confusion is coming from. It is standard notation but saying "set of distributions" makes it seem like it's multiple distributions. A state-action pair should map to *a* distribution over states.

---

> > > ### Author Response · Authors · 2024-11-28
> > > **Re Response and Revised Paper**
> > >
> > > We thank the reviewer for engaging in the discussion phase and we are happy that they better understand our contribution. We understand that the reviewer still has reservations about the writing and experimental evaluation. We have now uploaded a revised version of the paper, which beyond some minor changes to notation and grammatical improvements, includes an extended discussion on the practical issues with comparing our approach to LTL-based approaches directly (see Section 4.3). We have also extended the related work section to help better distinguish our work from similar approaches in the literature. Furthermore, In Appendix D we have provided a more extensive set of ablation studies, which includes testing our approach without counter factual experiences, and with potential-based reward shaping and cycle-based reward shaping (adapted from [2] to our purposes). We also provide additional results on the pacman environment from [1]. If the reviewer has further questions we are happy to answer them for the remainder of the discussion period. Also if the reviewer finds the revised paper more convincing, we encourage them to consider adjusting their score.
> > >
> > > [1] Policy Optimization with Linear Temporal Logic Constraints. Voloshin et. al. 2022.
> > > [2] LTL-Constrained Policy Optimization with Cycle Experience Replay. Shah et. al. 2024.

---

> ### Author Response · Authors · 2024-11-18
> **Re Questions**
>
> - How do different choices of reward shaping or learning for the safe policy based affect performance?
>
> One could consider using potential based reward-shaping for further improving the speed of convergence of the safe (backup) policy. We found that counter factual experience alleviated all issues we had with slow convergence for the safety properties that we considered in our experiments.
>
> - How complicated do the safety properties need to be for the approach to start breaking down?
>
> The only main restrictions on the complexity of the safety properties is whether the DFA can fit in memory, and how quickly counterfactual experiences can be run over the DFA. There is no restriction really for model checking as we leverage statistical model checking which computes the product states on-the-fly meaning we don't have to compute the full product Markov chain. For more complicated safety properties we might also want to increase the model checking horizon $N$ which will incur additional computation time although the dependence on $N$ is linear.
>
> - Have you tried comparing your approach to previous approaches to temporal-logic constrained policy optimization e.g. [1] and [2], rather than just CMDP-based approaches?
>
> We thank the reviewer for bringing these works to out attention, as both works seek to solves problems similar (but not identical) to our problem.  With regards to [1], [1] makes non-trivial assumptions, such as access to a generative model of the true dynamics $P$ from which they can sample transitions $s \sim P(s, a)$ for any state-action pair. This assumption immediately breaks the standard RL setting where we have to sequentially interact with the environment from a given initial state. Thus, setting up a fair comparison between our approach and [1] becomes unrealistic.
>
> The only assumptions made by [2] appear to be knowing the optimal discount $\gamma$ for the LTL-objective and choosing the Lagrange multiplier $\lambda > \lambda^*$ greater than the optimal $\lambda^*$. In this instance we could attempt to make a fair comparison, however [2] firstly lacks an open source implementation, and secondly we would need to take care to translate the DFA and its satisfaction condition to a corresponding Buchi automata, highlighting these technical details in the main paper might confuse readers and obfuscate our technical contributions.
>
> Some further discussion about [2], the objective function in [2] becomes $\pi^* = \arg\max_{\pi} [R_{\pi} + \lambda V_{\pi}]$ , the baseline (QL-cost) essentially optimizes this objective (without reward shaping, but with CFE) for a fixed $\lambda$ (in most of our experiments we choose $\lambda = 10.0$), in principle one could adaptively update $\lambda$ using the estimates of $R_{\textit{max}}$ and $R_{\textit{min}}$ during training, I think this is almost exactly what ROSARL is [3].
>
> [3] "ROSARL: Reward-Only Safe Reinforcement Learning" by Geraud Nangue Tasse, Tamlin Love, Mark Nemecek, Steven James, Benjamin Rosman
>
> Before the end of the discussion period we will try clear up any further questions you have, and we will aim to update the paper with additional ablations, although, owing to space constraints, these will likely have to be put in the appendix.

---

> > ### Comment · Reviewer_dvFE · 2024-11-26
> > **Thanks to the authors for their response Re: Questions**
> >
> > Thanks to the authors for their response to my questions.
> >
> > > We found that counter factual experience alleviated all issues we had with slow convergence for the safety properties that we considered in our experiments.
> >
> > This makes sense, thanks. It seems like counterfactual experience replay is probably most useful when the safety specifications are fairly simple and the resulting DFAs are parsimonious. Which is fine - most related works refer to similarly sized DFAs/safety specs. But CER has some scaling issues of its own, so it will be interesting to see how helpful CER will remain when the settings become more complicated.
> >
> > The other responses answer my questions, so thanks for that!

---

> ### Author Response · Authors · 2024-11-25
> **Remaining Issues**
>
> We beleive that most of the issues and reservations raised by the reviewer have been adressed, if the reviewer has no further comments we will deem these issues resolved and aim to add the additional ablations suggested by the reviewer to the revised version of our paper, before the end of the discussion period. In any case we encourage the reviewer to adjust their score as they see appropriate.

---

> ### Author Response · Authors · 2024-12-03
> **Remaining issues after revised paper**
>
> We gently encourage the reviewer to bring to our attention any remaining issues they have in light of the revised paper. If they find any of their issues resolved, for the most part, we encourage them to reconsider their score.

---

### Official Review · Reviewer_T6An · 2024-11-04

**Soundness:** 3
**Presentation:** 3
**Contribution:** 2
**Rating:** 5
**Confidence:** 4

**Summary:**

The paper presents an approach to ensure safety in reinforcement learning through a framework called lookahead shielding. This method dynamically identifies and avoids unsafe actions during both training and deployment by using a backup policy that activates whenever potential safety violations are detected. The authors validate their framework across various environments, showing it can achieve a balance between safety and performance, outperforming conventional constrained Markov Decision Process methods in some cases.

**Strengths:**

The paper presents an approach to safety in RL through its lookahead shielding framework, which requires minimal prior knowledge of the environment's dynamics. The framework dynamically assesses the safety of actions, adjusting in response to evolving environments. Extensive empirical validation in both tabular and deep RL environments demonstrates the method’s scalability and effectiveness in balancing reward and safety. Additionally, by comparing with constrained Markov Decision Process (CMDP) methods, the authors offer insights into the framework's performance relative to traditional approaches.

**Weaknesses:**

The framework seems to rely on a lookahead mechanism that verifies the probability of reaching an unsafe state after an action is selected, rather than ensuring safety beforehand. This approach implies that actions may be applied before verification results fully confirm safety, potentially allowing unsafe actions to affect the system before they are flagged. In contrast, a real-time synthesis approach would formally generate safe actions upfront, guaranteeing that no unsafe actions are applied. The lack of explicit real-time formal synthesis in the paper suggests that the framework provides posteriori (post-hoc) safety assurances rather than proactively constraining actions in real time. This could limit its effectiveness in safety-critical applications where preemptive safety guarantees are essential. For applications that require complete prevention of unsafe actions, a real-time synthesis approach might be more appropriate, as it directly generates actions that comply with safety constraints before execution. Am I missing something here?

Additionally, there is already a body of work on online shielding that the paper does not cite or compare with. Notable works include:

"Online Shielding for Stochastic Systems" by Bettina Könighofer, Julian Rudolf, Alexander Palmisano, Martin Tappler, and Roderick Bloem.

"Online Shielding for Reinforcement Learning" by Bettina Könighofer, Julian Rudolf, Alexander Palmisano, Martin Tappler, and Roderick Bloem.

"Safe Reinforcement Learning with Nonlinear Dynamics via Model Predictive Shielding" by Osbert Bastani.

**Questions:**

Could you provide an explanation of the novelty of your results in comparison to the following papers:

"Online Shielding for Stochastic Systems" by Bettina Könighofer, Julian Rudolf, Alexander Palmisano, Martin Tappler, and Roderick Bloem.

"Online Shielding for Reinforcement Learning" by Bettina Könighofer, Julian Rudolf, Alexander Palmisano, Martin Tappler, and Roderick Bloem.

"Safe Reinforcement Learning with Nonlinear Dynamics via Model Predictive Shielding" by Osbert Bastani.

---

> ### Author Response · Authors · 2024-11-18
> **Re Weaknesses**
>
> We thank for the reviewer for taking the time to read our paper and we appreciate their assessment.
>
> *"The framework seems to rely on a lookahead mechanism that verifies the probability of reaching an unsafe state after an action is selected rather than ensuring safety beforehand."*
>
> This is not strictly correct, the lookahead mechanism verifies the probability of reaching an unsafe state given the action proposed by the task (reward) policy, the verification is done in the approximate model of the environment, not the real one. Thus the action proposed by the task policy, if unsafe, is not actually used in the environment, preventing the selection of unsafe actions before they are actually performed.
>
> When we lack a precise model of the MDP or a safety-relevant abstraction of the MDP, as it is the case in many real-world application, synthesis cannot be done beforehand and the safety of the system cannot be absolutely guaranteed. Still, we show how useful the construction of an approximate transition model can be provided.
>
> We would like to take this opportunity to re-iterate the contributions of our paper. Firstly, we assume minimal prior knowledge, this includes, no access to a model of the MDP or a safety-relevant abstraction of the MDP, and no access to a hand-crafted backup policy. Both the approximate dynamics model of the MDP and the backup policy are learned online, meaning indeed, we cannot immediately provide safety assurances, however, in the tabular setting we can provide probabilistic safety assurances given that each state action pair is visited at least $\mathcal{O}\left(\frac{N^2|\mathcal{S}|^2}{{\varepsilon'}^2} \log\left( \frac{|\mathcal{A}||\mathcal{S}|^2}{\delta'}\right)\right)$ times. We discuss in the paper that this bound is quite conservative but in practice we can do much better.
>
> While we understand that this is not required, we also encourage the reviewer to read Appendix C, where we ablate our approach, by providing the model checker with a perfect model of the MDP and a perfect backup policy (computed with value iteration before training). In this instance we can immediately provide safety assurances before training, we also demonstrate that our approach (QL-Shield) quickly matches the performance of this ablation (QL-Exact) demonstrating that the dynamics model and backup policy have successfully been learnt.

---

> ### Author Response · Authors · 2024-11-18
> **Re Questions**
>
> Both [1] and [2] are closely related to "Safe reinforcement learning via probabilistic shields." by Jansen, Nils and K{\"o}nighofer, Bettina and Junges, Sebastian and Serban, Alexandru C and Bloem, Roderick
>
> We will comment on the differences between our approach and this body of work more generally. [1,2] are closely related to our approach in that verification or shielding of the agents actions is done online in real time, rather than exhaustively over the state space before training. [1,2,3] also consider a $k$-step lookahead horizon for model checking in the same way that we do, however there are several key differences between our approach and [1,2,3] which we highlight here. Firstly, [1,2,3] assume access to the precise transition dynamics of the MDP or at the very least a safety-relevant fragment of the MDP which must only be computed from the true MDP using prior knowledge, I quote them here: "In this paper, we consider the setting where the safety-relevant fragment of the MDP together with a temporal logic safety specification is given" [2]. We make no such assumption -- learning the dynamics of the MDP from scratch. [1,2,3] also do not provide any global safety guarantees in line with ours, I quote: "In contrast, a sequence of bad choices may violate $\varphi$ with high probability" [3], loosely speaking this is because verifying the safety of an action for $k$-steps does not guarantee that [1,2,3] don't go down a suboptimal path that violates the safety-requirement $\varphi$ with high-probability, unless we make additional assumptions. We address this issue by making the suitable assumptions on the model checking horizon (see Assumption 3.8 (line 330) in our paper). Furthermore, [1,2,3] also rely on planning in the reachable sub-MDP to compute safe actions, we do not need to plan rather the safe actions are learned and encoded in the safe (backup) policy, side-stepping the need for planning -- making our approach more practical in online scenarios.
>
> [1] "Online Shielding for Stochastic Systems" by Bettina Könighofer, Julian Rudolf, Alexander Palmisano, Martin Tappler, and Roderick Bloem.
>
> [2] "Online Shielding for Reinforcement Learning" by Bettina Könighofer, Julian Rudolf, Alexander Palmisano, Martin Tappler, and Roderick Bloem.
>
> [3] "Safe reinforcement learning via probabilistic shields." by Jansen, Nils and K{\"o}nighofer, Bettina and Junges, Sebastian and Serban, Alexandru C and Bloem, Roderick
>
> We now comment on the differences with [4]. [4] is specifically concerned with continuous state action spaces which are typically more tricky. However, in [4] the authors assume access to the transition dynamics, I quote "we focus on planning, where the dynamics are known", the authors also assume access to a backup policy which is computed with LQR and verified before training. No such assumption is made here. The environments considered in [4] also have equilibrium points in which the agent can remain in indefinitely given some control $u^*$, for discrete state spaces this corresponds to having safe end-components where the agent can remain in indefinitely, safe end-components do not always exist and are not trivial to compute.
>
> [4] "Safe Reinforcement Learning with Nonlinear Dynamics via Model Predictive Shielding" by Osbert Bastani.
>
> Summing up, we appreciate the reviewer for bringing these works to our attention. We will aim to include a similar (more concise) discussion in the revised version of our paper.
>
> In general [1,2,3,4] make non-trivial and often restrictive assumptions which we do not make, this makes it hard to setup a fair comparison between our approach and these works, rather we opt for a comparison with CMDP-based approaches that also make no assumptions about the MDP or backup policy.
>
> For the time being we invite the reviewer to ask any more questions they may have.

---

> ### Author Response · Authors · 2024-11-25
> **Remaining Issues**
>
> If the reviewer has no further comments we will deem the issues raised by the reviewer as resolved and aim to implement the changes mentioned above to the revised version of our paper before the end of the discussion period. In any case we encourage the reviewer to adjust their score as they see appropriate.

---

> ### Author Response · Authors · 2024-11-28
> **Revised Paper**
>
> We have now uploaded a revised version of the paper. Beyond some minor changes to notation, we have also extended the related work section to include the papers brought to our attention by the reviewer, and we have added additional experiments to the appendix. We kindly invite the reviewer to read the new material that they find interesting, and we invite them to ask any further questions they may have or adjust their score as they see appropriate.

---

> ### Author Response · Authors · 2024-12-03
> **Remaining issues after revised paper**
>
> We gently encourage the reviewer to bring to our attention any remaining issues they have in light of the revised paper and invite them to ask further questions which hopefully we can adress with the time left. If they find any of their issues resolved, for the most part, we encourage them to reconsider their score.

---

### Meta-Review · Area_Chair_9C2z · 2024-12-20

**Metareview:**

This paper introduces an online shielding method for reinforcement learning. In essence, during RL, and depending on what information has been learned/discovered already, potentially unsafe actions can be shielded (blocked). For cases where not enough information is available, a safe backup policy is trained.

The reviewers agree that the paper has a decent contribution, but that the contribution in the end it not too strong compared to previous approaches. The concept of 'minimal requirements' is slightly oversold. In the end, all real safety guarantees depend on a model that has to be learned. Other shielding approaches can also learn sufficient information and then shield safely. Moreover, it is somewhat hidden that the approaches largely rely on statistical model checking, which adds another level of  (only) statistical guarantees (on top of the already learned model).

**Additional Comments On Reviewer Discussion:**

The authors significantly improved their paper after the discussion, which led to partially better understanding and framing of the paper.

---

### Decision · Program_Chairs · 2025-01-22

Reject